# Prior preferences beneficially influence social and non-social learning

Tor Tarantola [1], Dharshan Kumaran[2], Peter Dayan [3] & Benedetto De Martino [2]

Our personal preferences affect a broad array of social behaviors. This includes the way we learn the preferences of others, an ability that often relies on limited or ambiguous information. Here we report an egocentric influence on this type of social learning that is reflected in both performance and response times. Using computational models that combine inter-trial learning and intra-trial choice, we find transient effects of participants' preferences on the learning process, through the influence of priors, and persistent effects on the choice process. A second experiment shows that these effects generalize to non-social learning, though participants in the social learning experiment appeared to additionally benefit by using their knowledge about the popularity of certain preferences. We further find that the domain-general egocentric influences we identify can yield performance advantages in uncertain environments.

---

[1] Department of Psychology, University of Cambridge, Downing Street, Cambridge CB2 3EB, UK. [2] Institute of Cognitive Neuroscience, University College London, 17 Queen Square, London WC1N 3AR, UK. [3] Gatsby Computational Neuroscience Unit, University College London, 25 Howland Street, London W1T 4JG, UK. Correspondence and requests for materials should be addressed to T.T. (email: tor.tarantola@gmail.com) or to B.D.M. (email: benedettodemartino@gmail.com)

Prior information can be useful when learning to navigate new environments[1-3]. The same holds true for social environments, which require us to learn and predict others' preferences, often based on limited information[4-7]. The consequences of poor predictions can be damaging—for example, to interpersonal relationships, businesses forecasting market trends, or governments attempting to resolve conflicts—so it is important to start with as much information as possible. In such cases, a useful starting point might be our own preferences—absent evidence to the contrary, it is reasonable to assume that other people prefer the same things that we do. Indeed, research in psychology has demonstrated that people tend to project their own values, traits, and preferences onto others[4, 5, 8, 9] and use themselves as priors when learning others' preferences[10, 11].

As we gather more information, we ought to update our predictions. Nevertheless, previous work has shown that egocentric influences tend to persist, even in the face of countervailing evidence[10, 12, 13]. This is particularly true when obtaining that evidence requires effort[14]. Still unclear is whether these influences can be overcome by enough evidence—that is, whether they merely act as priors in a learning process, or whether they persistently bias the intra-trial choice process. Also unknown is whether these influences are exclusively social or instead result from domain-general biases, and whether they help or hurt people's ability to make accurate predictions.

We investigated these questions by studying how participants learned which foods another person preferred. We found that participants' performance was strongly influenced by their own preferences (elicited beforehand), especially at the beginning of learning. However, even after learning had plateaued, participants continued to make more errors when the other person's preferences differed from their own. In a follow-up version of the experiment, a different group of participants performed exactly the same task, but were not told that the food items to be learned were the preferences of another person. This second, non-social experiment showed the influence of participants' preferences to be domain-general and potentially

applicable to reward learning more broadly. However, we also identified a key distinction between the social and non-social groups: participants in the social experiment appeared to use some knowledge about the popularity of the snack items, which improved their initial performance relative to the non-social group.

We compared four different computational models in order to isolate the relative influences of participants' preferences on the learning and choice processes. Following recent work[15], these models fed values learned over several trials into a drift diffusion model (DDM)[16-18], which describes the sequential sampling of noisy evidence within each trial. This allowed us to use both choice and response time data to evaluate whether preferences influenced inter-trial learning, the intra-trial choice process, or both. The most predictive model showed that both priors and *a priori* choice biases were influenced by participants' preferences.

Finally, we conducted a series of simulations to quantify how these influences affect expected performance. We compared the average performance of artificial actors who were influenced by their own preferences to neutral actors who were not. We found that the influences we observed in our experimental data—on both priors and choices—resulted in consistently higher performance on average, suggesting that they may actually be advantageous.

## Results

**Behavior.** Hungry participants were first asked to express their own preferences for food snacks using both a bidding procedure and a two-alternative forced choice task. They then performed a learning task in which they were instructed to learn the snack choices made by a randomly assigned pilot participant (their 'partner') between the same pairs of food items. They were told that, after making a response, a yellow feedback box would indicate the correct answer with 80% probability or the incorrect answer with 20% probability (Fig. 1). Participants were paid an additional £0.01 for every correct response.

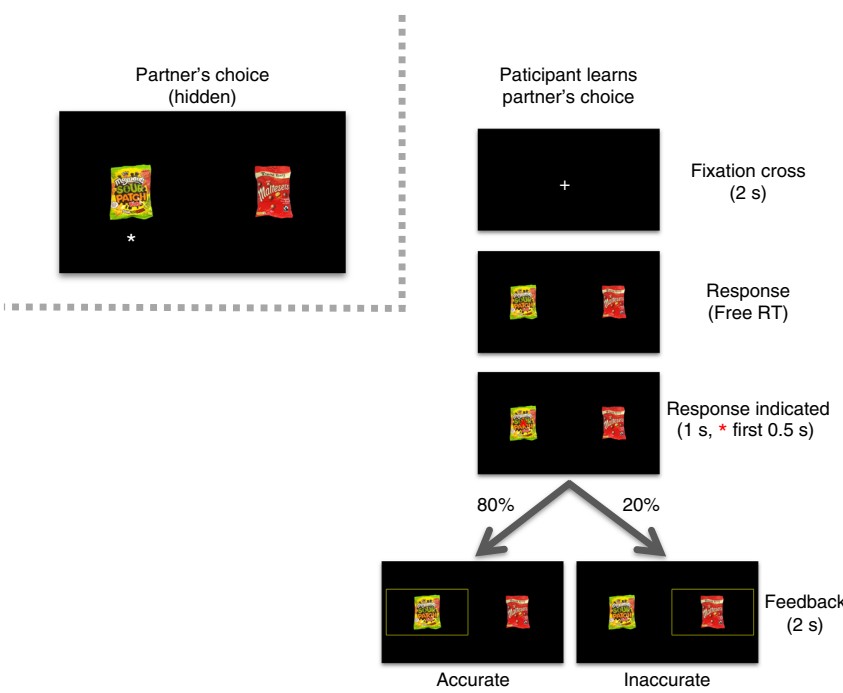

**Fig. 1** Learning task. Participants were asked to indicate which choice they believed their partner made. After making a response, a yellow feedback box indicated the correct answer with 80% probability. Participants saw 20 different pairs, interleaved, 30 times each for a total of 600 trials

On average, participants performed worse when learning their partners' preferences that differed from their own compared with preferences that they shared (Fig. 2a, top left). A mixed-effects logistic regression analysis showed that preference congruence—defined as the difference in the participant's bid (elicited beforehand) for the correct versus incorrect item ($\Delta v$)—had a significantly positive effect on performance ($n = 18{,}600$ observations across 31 participants; coefficient (S.E.): 0.56 (0.04), $z = 13.9$, $P < 10^{-14}$ Bonferroni corrected for multiple comparisons; Supplementary Table 1). We re-ran this analysis separately for the first, second, and last 10 trials of each item pair, and $\Delta v$ continued to have a positive effect on performance in each case

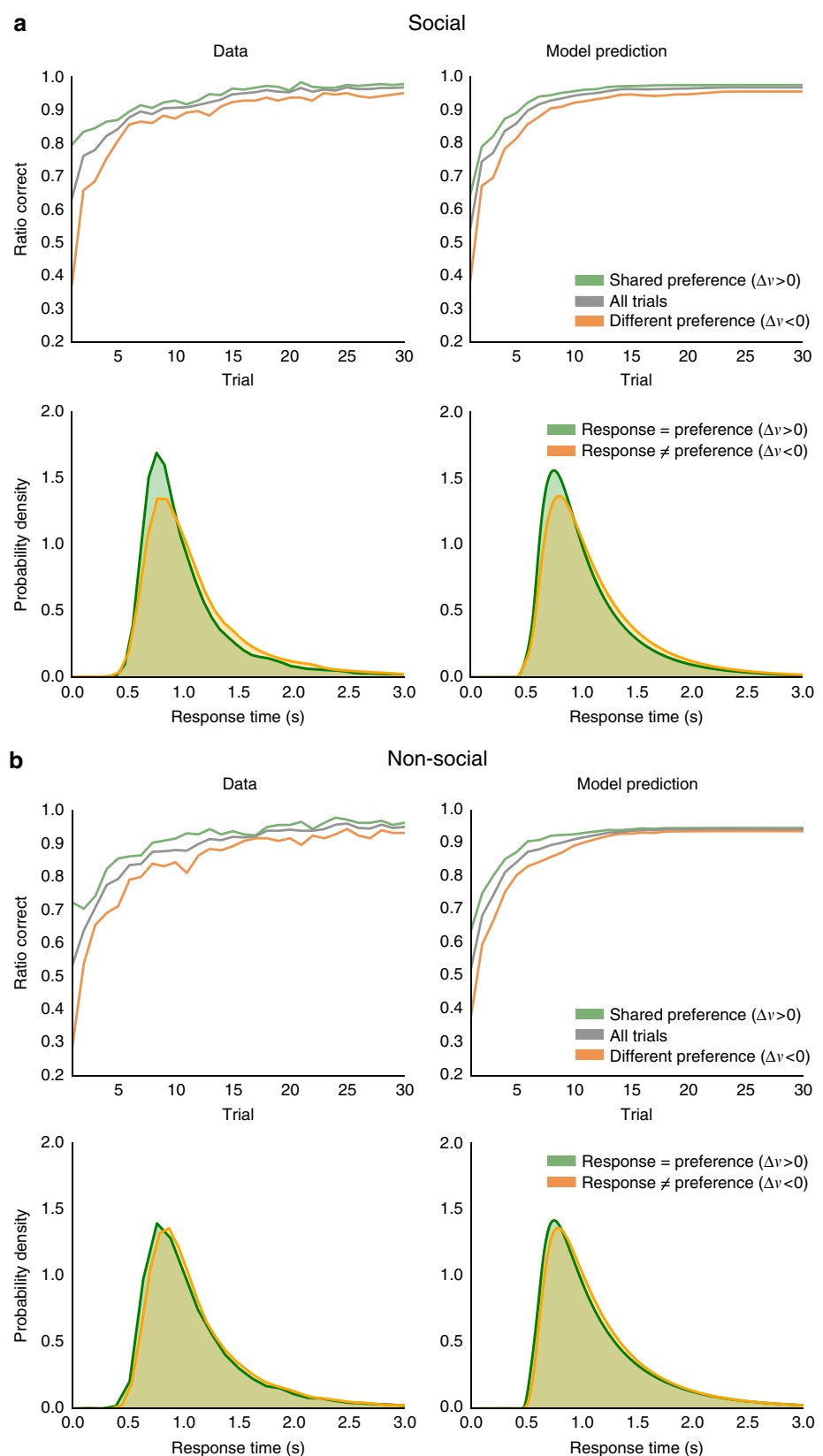

(each $n = 6{,}200$ observations; coefficients (S.E.): 0.42 (0.08) to 0.56 (0.05), each $z > 5.1$, each $P < 10^{-4}$ corrected; Supplementary Table 1). Notably, in the last 10 trials, there was no longer an effect of trial number on performance (coefficient (S.E.): 0.02 (0.02), $z = 1.0$, $P = 0.32$; Supplementary Table 1), although the effect of preference congruence persisted.

Participants were also faster making responses that matched their own preferences compared with responses that did not (Fig. 2a, bottom left). Response times were negatively predicted by the difference between the participant's bid for the item matching her response and her bid for its alternative (mixed-effects linear regression model predicting $z$-scored response times; $n = 18{,}596$ observations across 31 participants; coefficient (S.E.): $-0.12$ (0.01), $t = -12.3$; $\chi^2$-test of log-likelihood improvement from including the variable; $\chi^2(1) = 151.0$, $P < 10^{-14}$ corrected; Supplementary Table 2). This was also true for the first ($n = 6{,}199$), middle ($n = 6{,}198$), and last 10 trials ($n = 6{,}199$) of each item pair when analyzed separately with the same response time model (coefficients (S.E.): $-0.09$ (0.01) to $-0.21$ (0.02), each $t < -7.2$; each $\chi^2(1) > 51.1$, each $P < 10^{-10}$ corrected; Supplementary Table 2).

We also ran these performance and response time models using participants' own choices between the items (rather than the differences in their bids for each item, $\Delta v$) as a measure of preference congruence. These analyses yielded similar results for both performance (coefficients (S.E.): 0.36 (0.06) to 0.56 (0.04), each $z > 6.2$, each $P < 10^{-7}$ corrected; Supplementary Table 3) and response times (coefficients (S.E.): $-0.06$ (0.01) to $-0.15$ (0.02), each $t < -5.3$; each $\chi^2(1) > 28.0$, each $P < 10^{-5}$ corrected; Supplementary Table 4). Our task pseudorandomized both the number of trials between subsequent presentations of the same item pairs and the accuracy of the feedback presented on any given trial (see Methods). Nevertheless, to rule out potential confounds, we re-ran each of these regression models, controlling for these factors, on all but the first trials for each item pair. Specifically, we added as regressors (1) the accuracy of the previous feedback presented for the item pair and (2) the number of trials between the current and previous presentation of the item pair. These models continued to show significant positive effects of $\Delta v$ and participants' choices on performance (each $z > 5.1$, each $P < 10^{-4}$ corrected; Supplementary Tables 5 and 7) and significant negative effects on response times (each $t < -5.4$; each $\chi^2(1) > 28.8$, each $P < 10^{-5}$ corrected; Supplementary Tables 6 and 8). We also ran separate regression models to ensure that neither of these factors was predicted by $\Delta v$ (logistic model regressing previous feedback accuracy on $\Delta v$; $n = 17{,}980$ observations across 31 participants; coefficient (S.E.): 0.01 (0.03), $z = 0.3$, $P = 0.73$; linear model regressing gap between pair presentations on $\Delta v$; $n = 17{,}980$; coefficient (S.E.): 0.06 (0.17), $t = 0.4$, $P = 0.71$; Supplementary Table 17).

**Social versus non-social framing.** To determine whether the effect we observed was specific to the social nature of the task, we ran a second, non-social experiment in which participants performed the same task but received different instructions. In this second experiment, a different group of participants was asked to learn the correct item in each of the same 20 pairs of snacks. Unlike in the social experiment, these participants were not informed that the items they were learning were choices made by another person (see Supplementary Methods for instructions). Other than the difference in instructions, the two tasks were the same. As in the first, social experiment, participants in the second, non-social experiment were actually learning choices made by a randomly chosen pilot participant. Unlike in the social experiment, however, they were not told that they were learning another person's choices, but merely a set of correct answers.

Participants in the non-social experiment exhibited similar effects of preference congruence (Fig. 2b). As in the social experiment, $\Delta v$ had a significant positive effect on performance (mixed-effects logistic regression; $n = 18{,}000$ observations across 30 participants; coefficient (S.E.): 0.44 (0.03), $z = 13.1$, $P < 10^{-14}$ corrected; Supplementary Table 9) and a significant negative effect on response times (mixed-effects linear regression model predicting $z$-scored response times; $n = 17{,}999$ observations across 30 participants; coefficient (S.E.): $-0.08$ (0.01), $t = -8.7$; $\chi^2$-test of log-likelihood improvement from including the variable; $\chi^2(1) = 74.6$, $P < 10^{-14}$ corrected; Supplementary Table 10). As in the social experiment, these effects were still significant (each $P < 0.02$ corrected) when substituting choices for $\Delta v$ (Supplementary Tables 11, 12, 15, and 16), when controlling for previous feedback accuracy and trials since last pair presentation (Supplementary Tables 13–16), and when analyzing the first, second, and last 10 trials separately (Supplementary Tables 9–16).

However, we did note an important difference between the social and non-social groups: participants in the social group performed significantly better on the first trial for each item pair, before any feedback had been presented, compared to participants in the non-social group (mixed-effects logistic regression predicting first-trial performance, regressing on $\Delta v$ and group (social/non-social); $n = 1{,}220$ observations across 61 participants; coefficient (S.E.) of social group variable: 0.38 (0.13), $z = 2.9$, $P = 0.003$; Supplementary Table 18). When controlling for $\Delta v$, participants in the non-social group performed no better than chance (coefficient (S.E.) of intercept: $-0.001$ (0.09), $z = -0.01$, $P = 0.99$; Supplementary Table 18). This difference was not due to different effects of $\Delta v$ on first-trial performance, since this effect did not differ significantly between groups when an interaction term was added to the model (coefficient (S.E.) of $\Delta v \times$ group interaction: 0.17 (0.23), $z = 0.8$, $P = 0.45$; Supplementary Table 18). Rather, we hypothesized that this difference could be due to some knowledge that participants in the social group used about the idiosyncrasies of their own preferences relative to the general population, allowing them to adjust their responses to match the latter rather than the former.

---

**Fig. 2** Observed and predicted performance and response times. **a** In the social group ($n = 31$), participants made more errors on trials in which their preferences differed from that of their partner (*orange*, $n = 6{,}660$ observations) compared with pairs in which they shared their partner's preference (*green*, $n = 10{,}830$ observations). This difference was especially pronounced early in the task but persisted over the course of learning. They were also faster making responses that matched their own preference (*green*, $n = 10{,}962$ observations) compared with responses that did not (*orange*, $n = 6{,}524$ observations). **b** In the non-social group ($n = 30$), participants exhibited similar effects of preference congruence on performance (congruent, *green*, $n = 9{,}510$ observations; incongruent, *orange*, $n = 7{,}470$ observations) and response times (congruent, *green*, $n = 9{,}868$ observations; incongruent, *orange*, $n = 7{,}111$ observations), but performed significantly worse on first trials compared to participants in the social group. (See main text for statistical tests.) Model predictions for both groups are from the dual influence model with item popularity. For performance, preference congruence was indexed by the difference in the participant's bid for the correct item minus her bid for the incorrect item ($\Delta v$). For response times, $\Delta v$ was the participant's bid for the item matching her response minus her bid for its alternative. Performance is averaged across participants, and response times are averaged across trials and participants. Distributions are kernel density smoothed

To test this hypothesis, we regressed first-trial performance on both $\Delta v$ and item popularity. We indexed item popularity using the percentage of other participants in both experiments who bid more for an item compared to its alternative (Supplementary Table 19).

Note that the level of agreement varied among item pairs: 90% of participants bid more for the Twix bar than for the Polo Fruits candies, but participants were about evenly split between the Bounty bar and sweetcorn; the mean level of agreement was 69.8%.

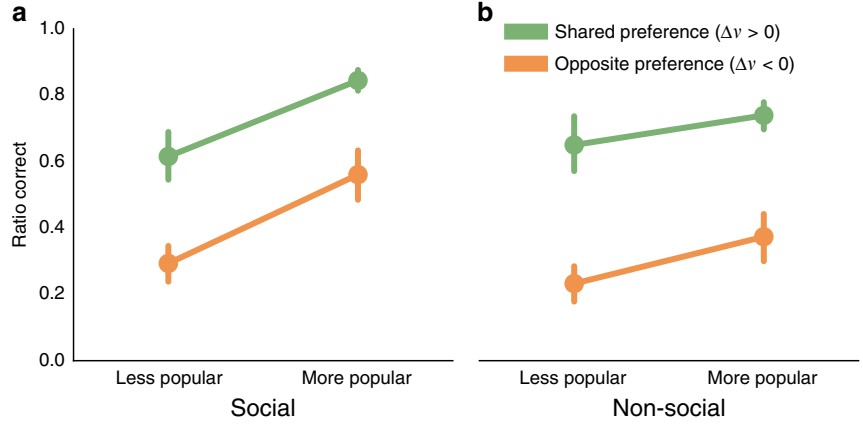

**Fig. 3** Performance on first trials for each item pair, separated by preference congruence and item popularity. **a** In the social experiment ($n = 31$), participants performed significantly better on first trials when they shared their partner's preference (*green*, $n = 361$ observations) than when they did not (*orange*, $n = 222$ observations). They also performed better when the correct answer was relatively more popular ($n = 372$ observations) than when it was relatively less popular ($n = 223$ observations). **b** The first-trial performance of participants in the non-social experiment, while better for shared (*green*, $n = 317$ observations) than for unshared preferences (*orange*, $n = 249$ observations), was less sensitive to the item's popularity (more popular, $n = 365$ observations; less popular, $n = 211$ observations; see main text for statistical tests.) 'More popular' items are those preferred by more than 50% of participants in both experiments; 'less popular' items were preferred by fewer than 50%. Error bars represent bootstrapped standard errors (ranging between 0.03 and 0.08) clustered by participant

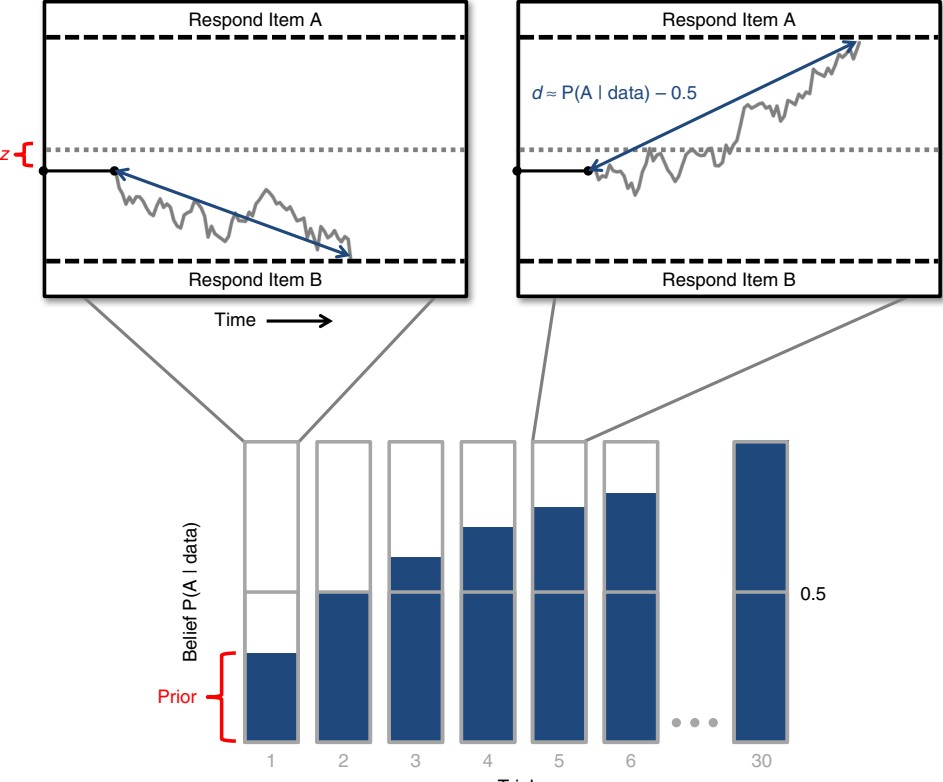

**Fig. 4** Schematic representation of two-level model of information processing. The belief about the probability of an option being correct increases as the participant sees more feedback over the course of several trials (*bottom*). On each individual trial, this probability is converted into a choice by setting the drift rate in the diffusion model (*top*). In this model, there are two potential points at which the participant's preference could influence her behavior: the prior in the learning process, which affects the drift rate, and the bias in the starting point of the choice process, both highlighted in *red*. We model these influences as softmax functions of the difference in the participant's bids for the correct and incorrect items ($\Delta v$), with inverse temperatures $\beta_{\Delta v}$ (prior) and $\kappa$ (choice bias)

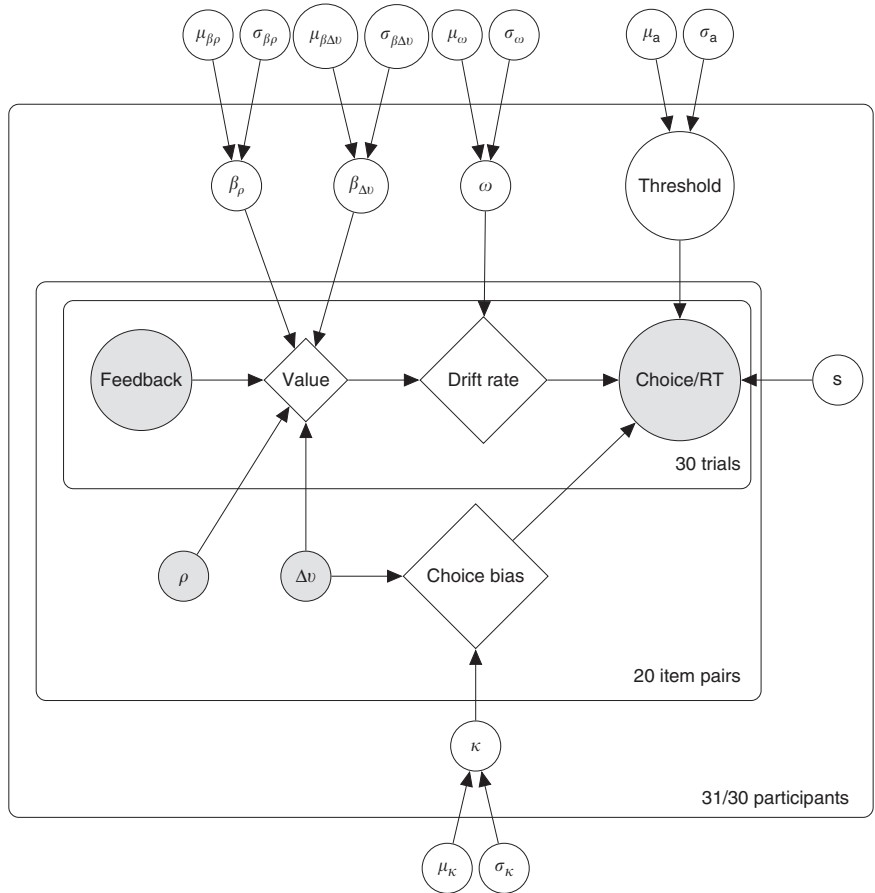

**Fig. 5** Graphical schematic representation of the dual influence model with item popularity. Unshaded circular nodes represent unobserved latent parameters that were estimated in the model. Diamond nodes represent variables whose values are fully determined by other nodes, and shaded nodes represent observed data. Except for non-decision time, latent parameters were estimated hierarchically, which assumed that each participant's latent parameter value was the product of a group-level distribution with mean $\mu$ and standard deviation $\sigma$. $\beta_\rho$ is the influence of item popularity on the prior; $\beta_{\Delta v}$ is the influence of preference on the prior; $\omega$ is the weighting term for translating learned values into a drift rate; $\rho$ is the relative popularity of the item; $\Delta v$ is the participant's relative preference for the item; $\kappa$ is the influence of preference on choice bias; and $s$ is the non-decision time

Including item popularity significantly improved a logistic model of first-trial performance in the social group (mixed-effects logistic model regressing on $\Delta v$ and item popularity; $n = 620$ observations across 31 participants; $\chi^2$-test of log-likelihood improvement from including item popularity; $\chi^2(1) = 38.3$, $P < 10^{-8}$ corrected; Supplementary Table 18) and, to a lesser extent, in the non-social group ($n = 600$ observations across 30 participants; $\chi^2(1) = 8.1$, $P < 0.01$ corrected; Supplementary Table 18). This effect was significantly greater for the social group than the non-social group ($\chi^2$-test of log-likelihood improvement from including group × item popularity interaction in a combined model; $n = 1{,}220$ observations across 61 participants; $\chi^2(1) = 5.7$, $P = 0.017$; Supplementary Table 18; illustrated in Fig. 3 and Supplementary Fig. 1).

We also noted that responses for the first trials were significantly faster in the non-social group than in the social group (mixed-effects linear model regressing on $\Delta v$ and group (social/non-social); $n = 1220$ observations across 61 participants; coefficient (S.E.) of social group: 0.65 (0.15), $t = 4.4$; $\chi^2$-test of log-likelihood improvement from including group; $\chi^2(1) = 17.6$, $P < 10^{-4}$ corrected; Supplementary Table 20), but were not significantly different for other trials ($n = 35{,}375$ observations across 61 participants; coefficient (S.E.): $-0.02$ (0.01), $t = -1.4$; $\chi^2(1) = 2.1$, $P = 0.15$; Supplementary Table 20). This suggests that, absent other information, participants in the social group may have been more deliberate in their first responses than

participants in the non-social group, possibly from reasoning about their partners' likely preferences as potentially distinct from their own.

**Computational models**. To test how the effects we observed arise algorithmically, we constructed a series of computational models that combine two levels of information processing: the inter-trial learning process; and the intra-trial choice process, which converts learned values into responses during noisy evidence accumulation (Fig. 4). At the inter-trial level, we characterized beliefs as being those of an ideal Bayesian observer that infers the probability of an option being correct given the cumulative feedback observed, having been told the probability of the feedback on each trial being correct (see Methods). We also tested versions of our models using Rescorla-Wagner-type learning rules, which achieved comparable results (Supplementary Fig. 2).

We modeled the intra-trial choice process using the DDM, which describes the noisy accumulation of evidence leading up to a single response. The DDM imagines a single decision particle drifting toward one of two response thresholds at an average rate proportional to the relative strength of evidence for each response. When the particle reaches a threshold, the participant makes the associated response. The trajectory of this particle is subject to Gaussian white noise, which leads to errors and variation in response times. As the strength of evidence increases—for example,

over the course of learning—the effect of this noise relative to the evidence decreases, leading to fewer errors and faster responses. A key advantage of the DDM is that it allows us to model both choices and response times at the level of a single trial.

To locate the source of the behavioral influence of participants' preferences, we tested four alternative models: (1) an influenced prior model, in which the drift process has a neutral starting point, but the prior associated with the Bayesian beliefs (which influences the drift rate—see Fig. 4) is a softmax function of the participant's relative preference for the correct versus incorrect item, $\Delta v$, with inverse temperature $\beta_{\Delta v}$. We define the prior belief at trial $n=1$ for an item pair, before any evidence has been received, as

$$P(A)_{n=1} = \frac{1}{1 + e^{-\beta_{\Delta v}\Delta v}} \qquad (1)$$

where $A$ is the correct item. These probabilities are then updated with additional feedback and influence the drift rate for each trial.

(2) An influenced choice model in which the priors are neutral but the DDM's starting point is biased. We define this bias as the portion, $z$, of the distance between the lower and upper thresholds, modeled as a softmax function of $\Delta v$ with inverse temperature $\kappa$:

$$z = \frac{1}{1 + e^{-\kappa\Delta v}} \qquad (2)$$

This model assumes that participants are optimal and neutral in their learning, but are biased in how they convert that learning into a response during intra-trial evidence accumulation.

(3) A dual influence model, in which both priors and choices are influenced by the participant's preferences.

(4) We compared these three models to a fourth, neutral model, in which neither the prior nor the choice bias is influenced by the participant's preferences.

For each of these models, the drift rate, $d$, at a particular item pair's trial, $n$, is specified as the difference between the probabilities of each item being correct given the feedback data accumulated for that item pair up to that point:

$$d_n = \omega(P(A|data_n) - P(B|data_n)) \qquad (3)$$

$A$ is the correct item, $B$ is the incorrect item and $\omega$ is a weighting term that translates probabilities into drift rates.

We fitted these models to both choices and response times using fully Bayesian parameter estimation[19, 20], which returns the most likely parameter estimates as well as the uncertainty around those estimates. All models were hierarchical (see Methods; Fig. 5). We then used leave-one-out cross-validation[21, 22] to determine which model had the best estimated predictive accuracy, measured by its expected log pointwise predictive density (ELPD) for a new dataset (see Methods). This out-of-sample validation method naturally accounts for overfitting and includes a measure of uncertainty, as indicated by the S.E. of the ELPD. For both social and non-social groups, we found that the dual influence model had the best predictive accuracy of the four models (social/non-social: ELPD=−8,077.8/−11,141.8, S.E.=240.0/195.9; Fig. 6a), significantly outperforming the second-best, influenced choice model (ELPD difference = 82.1/117.2, S.E. = 19.4/20.4).

To model the effect of item popularity that we identified in the behavioral analysis, we incorporated an index of the relative popularity of the items ($\rho$) into the prior of the dual influence model, weighted by inverse temperature $\beta_{\rho}$, to test whether this further improved the model's predictive accuracy for either the social or non-social contexts. We defined $\rho$ as the ratio of other participants who bid more for the correct item minus 0.5. (A $\rho$ of 0.25, for example, means that 75% of other participants preferred the correct item, while a $\rho$ of −0.25 means

that 25% of other participants preferred it; see Supplementary Table 19 for item pairs and their popularity.)

$$P(A)_{n=1} = \frac{1}{1 + e^{-(\beta_{\Delta v}\Delta v + \beta_{\rho}\rho)}} \qquad (4)$$

For the social group, this factor allows participants to incorporate their knowledge about the popularity of the different items into their priors. This should have no effect on the prior for a participant in the non-social group, who was unaware that the items being learned reflected the preferences of another person, and therefore would have no reason to consider their relative popularity.

Consistent with this hypothesis, the inclusion of item popularity significantly improved the predictive accuracy of the dual influence model for the social group (ELPD difference = 59.6, S.E. = 15.4), but not for the non-social group (ELPD difference = 17.5, S.E. = 10.8; Fig. 6a). For both groups, this model showed positive effects of preference ($\Delta v$) on both the DDM starting point and the priors for each item, as well as a positive effect of item popularity ($\rho$) on the priors for the social group (Fig. 6b; see Supplementary Figs 3 and 4 for additional parameter estimates).

The dual influence model with item popularity provided good descriptions of response time distributions, including the differences we observed between responses that matched and did not match participants' own preferences, for both social and non-social groups (Fig. 2, right panels). The model's predicted response time quartiles were all within 0.07 s of the actual data (Supplementary Table 21). The model also provided good descriptions of performance differences over the course of learning. We compared the mean model prediction to the actual ratio of correct responses for each item pair trial number. The mean difference between the prediction and the data was 0.01 (S.E.M. = 0.004), or one percentage point, for the social group, and 0.015 (S.E.M. = 0.004) for the non-social group. However, the model provides somewhat weaker fits for the first trials in each item pair (Supplementary Table 22), as the data seem to show an even greater effect of preference congruence at the beginning of learning than our model predicts. It is possible that the amount of noise in the choice process increases over time, perhaps due to fatigue or memory decay, leading our model to overestimate the amount of noise in earlier trials.

**Effects on performance**. We next asked whether this influence of preference might confer a performance advantage over a hypothetical actor who is uninfluenced by her own preferences. To test this, we performed a series of simulations across a range of different values for the influence of preference on choices ($\kappa$) and priors ($\beta_{\Delta v}$). These parameter values ranged from 0 to 8 times the actual group means recovered from the dual influence model fitted to the non-social group. We centered our simulations on the non-social group's parameter values in order to evaluate the efficiency of behavior in a more domain-general context, though the values themselves were very similar to those of the social group (Supplementary Figs 3 and 4). Using the preference data collected during both social and non-social experiments ($n=61$), we simulated each participant learning each other participant's preferences 1000 times (to take account of the probabilistic feedback), totaling 3,660,000 simulated experimental sessions for each combination of parameter values. We then compared the mean expected performance of each simulation against that of a Bayesian learner with a neutral prior and unbiased choice process.

Importantly, each simulation reflects the stochastic dynamics of the drift process by including an equal degree of noise in how

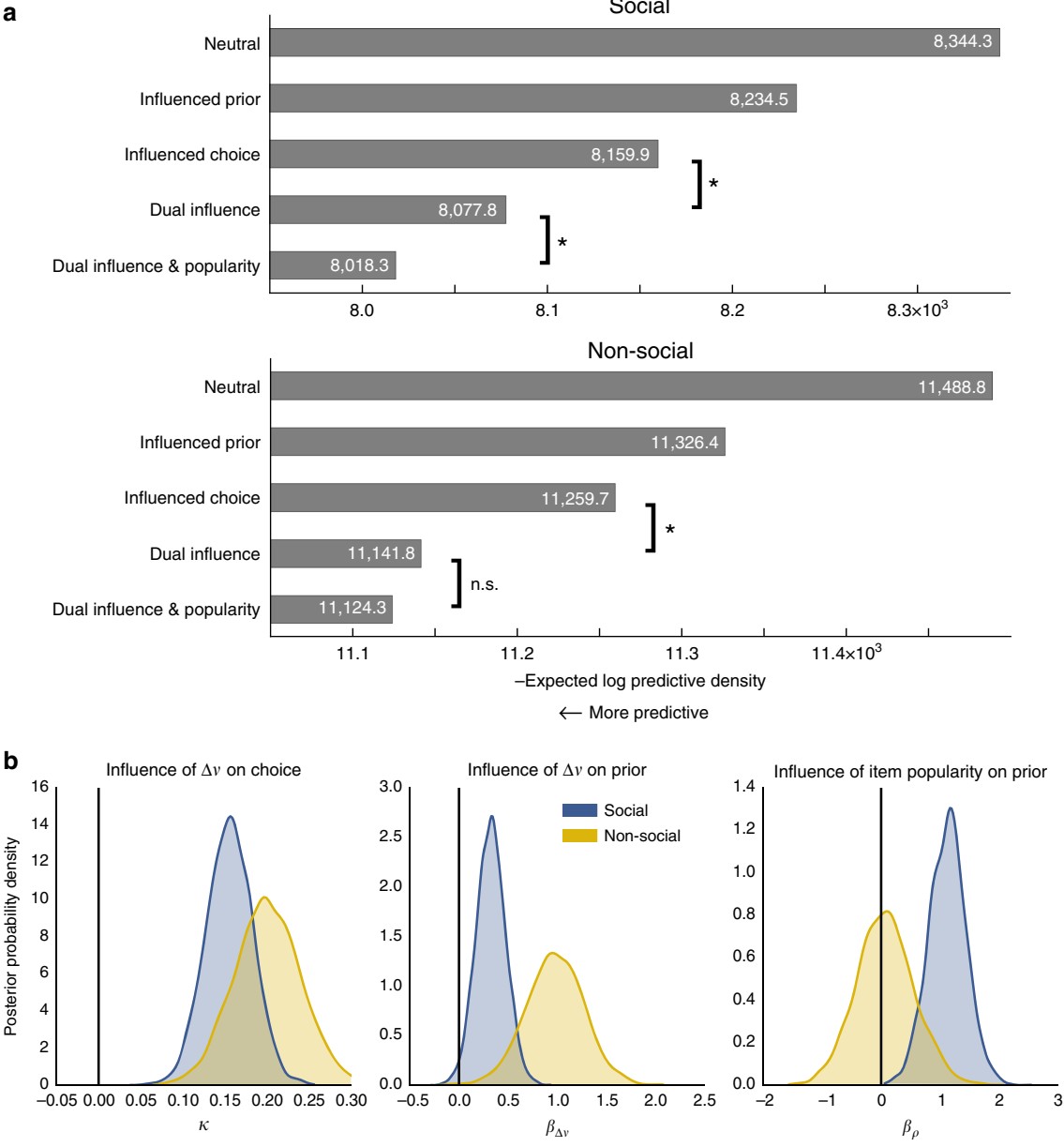

**Fig. 6** Computational model results. **a** Negative estimated log predictive density of the five models fitted to both social and non-social groups (lower means more predictive). * $P < 0.05$, two-tailed. **b** Group mean parameter estimates for the preference and item popularity weight terms in the dual influence model with item popularity, shown for both social (*blue*) and non-social (*gold*) groups. Values greater than 0 indicate a mean positive effect while values less than 0 would indicate a negative effect. Distributions show the uncertainty in the parameter estimates of the group means and are kernel density smoothed. While preference positively affects priors and choice processes in both groups, item popularity affects priors only in the social group

evidence is converted into responses, resulting in a predictable number of errors. While an optimal strategy would deterministically choose the option with greater evidence, no matter how scant, our simulations took the more realistic approach of assuming that noise from internal and external sources will tend to pollute this process.

On average, even without the benefit of knowledge about the items' relative popularity, the combination of an influenced prior and an influenced choice process performed the best overall. The best-performing dual influence strategy outperformed the neutral learner by a mean of 0.71 percentage points (Fig. 7a). By comparison, the best-performing influenced prior strategy achieved a mean advantage of 0.46 percentage points, and the best influenced choice strategy yielded a mean advantage of 0.38

percentage points. The dual influence strategy using the parameters recovered from our data outperformed the neutral learner by a mean of 0.69 percentage points. The best influenced prior strategy conferred an advantage chiefly at the beginning of learning, outperforming the neutral learner by 7.2 percentage points on the first trial. However, this advantage virtually disappeared once learning plateaued. On the other hand, the influenced choice strategy conferred a comparatively smaller advantage at the beginning of learning—1.2 percentage points on the first trial—but its advantage over the neutral actor persisted throughout the task, leveling off at 0.2 percentage points. This persistent advantage can be explained by the fact that the lapses (the cause of errors due to simulated noise in the choice process) were random in the neutral actor but biased toward the preferred

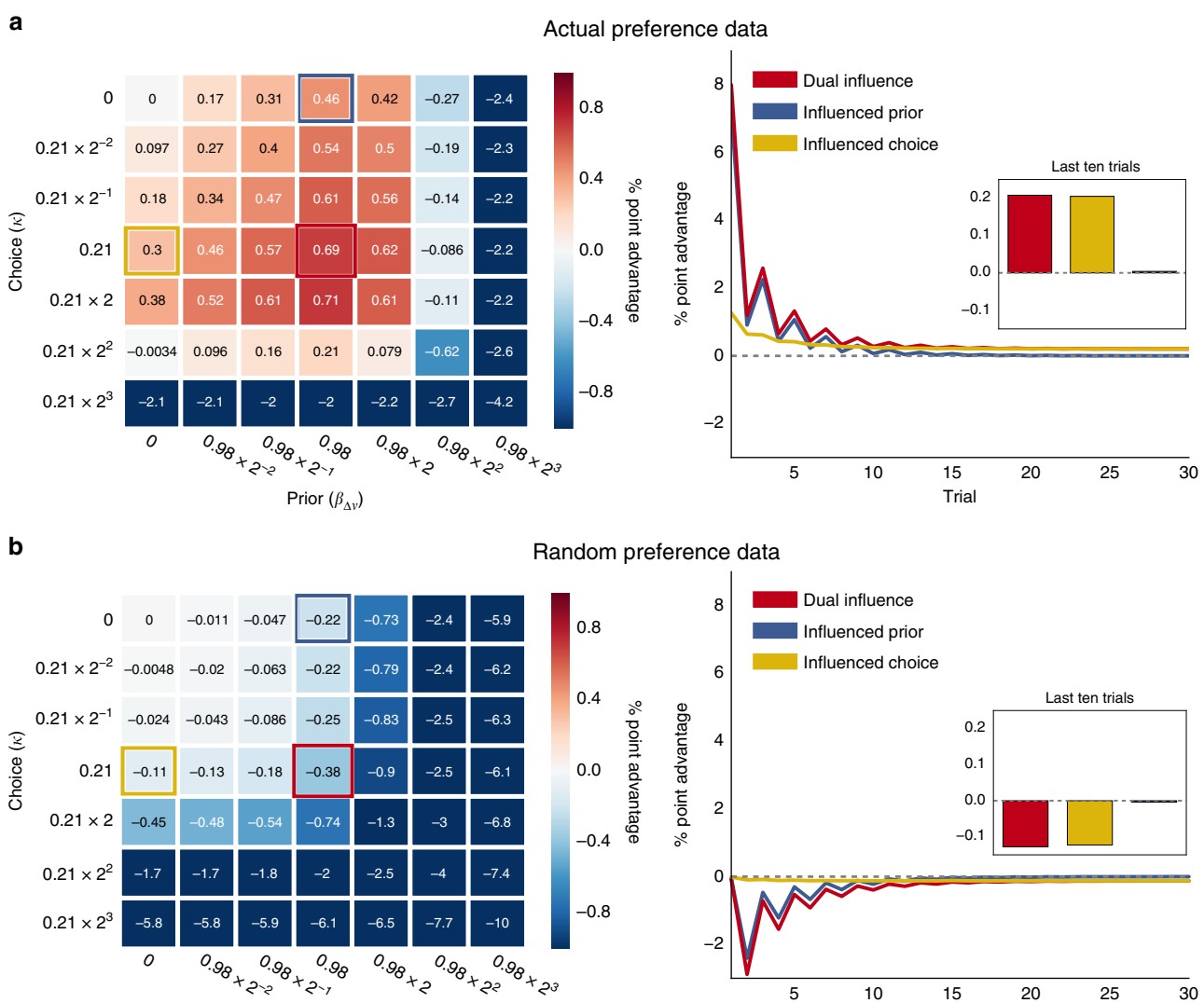

**Fig. 7** Simulated mean performance advantages conferred by different loci of preference influence. **a** A dual influence strategy, which combines influences of preference on both the prior and the choice bias, results in the greatest average performance in our simulated task using actual preference data. The heat map (*left*) shows the percentage point advantage over a neutral actor for different combinations of choice (vertical axis) and prior (horizontal axis) parameter values, centered at the actual means of the group mean parameter estimates from our dual influence model (highlighted in *red*), estimated from participants in the non-social group. This strategy outperforms both influenced choice (*yellow*) and influenced prior (*blue*) strategies. An influenced prior strategy confers a substantial performance advantage at the beginning of learning (*right*), but which disappears after learning has plateaued (inset). By contrast, an influenced choice strategy confers a smaller advantage at the beginning of learning, but it continues to slightly outperform a neutral actor even after learning has plateaued. The dual influence strategy combines these advantages for a performance boost not only at the beginning of learning but also after learning has plateaued, leading it to outperform the three other strategies. **b** When preference data are generated randomly, and therefore not predicted by a participant's own preferences, being influenced by one's own preferences results in a performance disadvantage compared to a neutral strategy. Performance is indicated in terms of the percentage point difference in correct responses between the indicated strategy and a neutral actor with identical drift weight ($\omega = 1.53$) and threshold ($a = 2.10$) parameters, which were set to the means of the group mean parameter estimates recovered from the dual influence model fitted to the non-social group's data

item in the influenced choice strategy. The dual influence strategy combined the advantages of both the influenced prior and influenced choice strategies and resulted in the greatest overall performance. We also found that this performance advantage was greater in simulations where we decreased the reliability of the feedback, hence delaying the influence of the evidence relative to other factors (Supplementary Fig. 5). When the probability of correct feedback was 0.7, the dual influence strategy using parameters recovered from our data yielded a mean advantage of 1.1 percentage points. This advantage increased to 1.8 percentage points when feedback reliability was 0.6. Because the balance of

these influences depends on the amount of evidence presented, the relative performance of different strategies would change depending on the number of trials.

The advantage we observed in our simulations resulted chiefly from the fact that participants' own preferences were, on average, predictive of their partners' preferences. Put another way, $\Delta\nu$ correlates strongly with the ratio of other participants who prefer the correct item (Pearson's $r = 0.48$, $P < 10^{-69}$). It is possible, therefore, that being influenced by one's own preferences—or, more generally, by experience of a stimulus's value in a previous context—might be maladaptive where those values do not

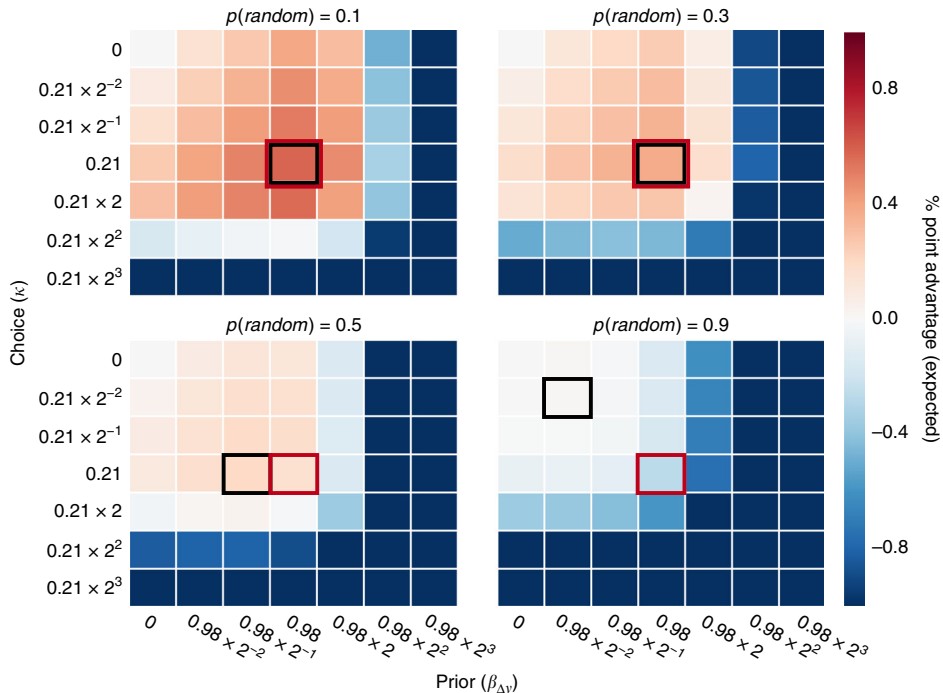

**Fig. 8** Expected performance advantages for different probabilities of random preference data. When there is a less than certain probability that the data are random—and a non-zero probability that the data are correlated to the actual degree observed in the data—the expected performance advantage is nevertheless maximized by a dual influence strategy (parameter combinations yielding the highest performance are highlighted in *black*, the observed parameter combination in *red*). Of the parameter combinations we simulated, participants exhibited behavior in the highest expected performance range for values of *p(random)* between 0.05 and 0.37

transfer across contexts. To test this, we ran a second series of simulations (Fig. 7b) in which preference data were randomly generated, causing participants' preferences not to be predictive of those of their partners. In this case, being influenced by one's own preferences conferred a performance disadvantage compared to a neutral actor, although, for the parameter values we observed in our data, this disadvantage happened to be smaller than the advantage gained when the actual preference data were used.

In the natural world, there is some uncertainty about whether previously learned stimulus values are applicable in novel contexts. An efficient strategy would seek to maximize expected rewards by accounting for this uncertainty. We calculated these expected rewards for a range of uncertainty levels:

$$E = p(random) * V_{random} + (1 - p(random)) * V_{predictive} \quad (5)$$

where *E* is the expected value of a strategy, *p(random)* is the probability that rewards are random—that is, unpredicted by a stimulus's prior value—and *V* is the expected value of the strategy in each context. We found that the effect of preference on participants' actual behavior was remarkably efficient for most values of *p(random)* below 0.5 (Fig. 8). In fact, of the parameter combinations we simulated, participants' behavior yielded the highest expected percentage point advantage (between 0.30 and 0.64) for values of *p(random)* between 0.05 and 0.37.

**Discussion**

In this study, we offer a computational account of how people's prior preferences influence both social and non-social learning: by affecting priors in the learning process, and by persistently biasing the choice process, which converts learned values into responses. Recent neurocomputational accounts have similarly begun to illuminate the interactions between our own preferences

and those of others[23–28], showing that, in some cases, they may be represented by the same neural populations[25–27]. This previous work has generally explored these dynamics during social conformity, in which participants' subjective valuations are influenced by others' opinions[25, 26, 28–37] in a process that resembles Bayesian inference[28, 37]. Our study examines the converse relationship, in which our prior preferences shape our expectations of what others prefer—an effect often referred to in the psychological literature as social projection[4]. Previous research on social projection has found that people exhibit an egocentric bias, believing that others share their opinions, traits and preferences more than they actually do[4, 6, 8, 9, 14, 38]. People have also been shown to use themselves as priors when learning about others—an effect that can dissipate with additional evidence[10, 11]. Our dual influence model describes this effect as the product of two influences—on both priors and choices—each with qualitatively different implications.

In Bayesian terms, a biased prior describes an informed expectation in the absence of evidence[1–3]. Our participants expressed priors about each correct answer that were strongly influenced by their own preferences. As they received probabilistic feedback indicating the correct answer, participants updated their priors to improve their accuracy. In addition to their own preferences, participants in the social group—who knew they were learning the preferences of another person—appeared also to incorporate some knowledge about the items' popularity into their priors. This is consistent with previous work suggesting that people may exhibit less social projection when they recognize that their own preferences are atypical[4, 6, 7]. Compared to the non-social group, participants in the social experiment also performed significantly better, and more slowly, on first trials. This suggests that they may have responded more deliberately before any feedback had been presented, perhaps reasoning about

the relative popularity of different items—a consideration that would not have been useful in the non-social experiment. Nevertheless, participants in both social and non-social groups had priors that were similarly influenced by their own preferences, suggesting that these biases may result from a domain-general process by which learning is guided by prior experience of a stimulus's value in other contexts.

The influence we identified on the prior somewhat resembles a Pavlovian bias, which can develop when a stimulus previously associated with a reward elicits an automatic response during instrumental learning. Importantly, this bias can emerge despite the response itself having never been reinforced[39]. While they can be maladaptive in certain circumstances, Pavlovian biases can also prove advantageous in environments where the value of a stimulus is relatively stable across contexts (cf. ref. [40, 41]). For example, a foraging animal may learn to associate a certain stimulus—such as a unique birdsong—with the presence of berries. If the animal heard this birdsong every time it encountered a berry patch, it may be a reliable cue for the presence of food. Therefore, it would behoove the animal to approach areas from which the birdsong emanates, even though approaching the birdsong itself was never directly reinforced. Similarly, in our learning task, selecting a snack stimulus—an approach behavior —was partly elicited by that snack's *a priori* value to the participant.

We also observed a degree of behavioral variability in participants' responding, meaning that they continued to make some errors even after learning had plateaued. This type of variability is often attributed to internal noise in neural processes[42], which is accounted for in the DDM by assuming stochasticity in how external evidence translates into the decision particle's trajectory toward a threshold. Critically, in identifying a bias in the DDM's starting point, we found that the effect of this noise was biased in favor of the participant's preferred item. (See Methods for a more detailed discussion of the DDM's parameters and their effects on behavior.)

Both learning and choice influences proved to be advantageous. But why might relying on prior preferences confer an advantage? Psychological theories have proposed that social projection might be a key part of a rational induction process, by which one's own preferences act as a reliable indicator of what others are likely to prefer. Because the average person is more likely to hold a majority than a minority opinion, an egocentric bias can result in better average performance[4, 6]. Our simulations examined the extent to which different types and levels of preference influence would lead to performance advantages and how these advantages might change as additional evidence accumulates. We found that simulated participants whose priors and choice biases were both influenced by their own preferences would, on average, perform better in our task than an actor with a neutral prior and an unbiased choice process. While the relative advantages in our task were small, such a strategy could be adaptive when applied to the many choices faced by an animal over its lifetime. The advantages of both influences derived from the fact that participants' own preferences were, on average, predictive of the correct answer. In the case of the prior, the participants' preferences take the place of evidence until enough feedback has accumulated. In the case of the choice bias, however, this influence persists even in the face of substantial evidence to the contrary. This bias might still confer an advantage because of the noise inherent in the choice process, which causes behavioral variability. For the neutral actor, this variability favors neither the correct nor the incorrect answer. For the actor with a biased DDM starting point, however, this variability effectively favors the participant's preferred item. Because the participant's preferred item is, on average, more likely to be correct than incorrect, this bias improves

overall performance by reducing the number of errors due to internal noise.

Of course, there are some cases in which stimulus values may not transfer across contexts. While a birdsong may predict the presence of berries in the forest, approaching the birdsong on the prairie may have no instrumental value. In such a case, approaching a non-predictive stimulus may cost an animal a better foraging opportunity. When taking these potential costs into account, the values of our participants' prior and choice influence parameters were remarkably efficient compared to other possible value combinations.

These results point to several avenues for further research. While our paradigm was designed to be somewhat naturalistic— learning others' preferences often requires remembering discrete choices—learning about others can also involve added layers of complexity. For example, certain types of preference information might be useful for inferring more general features of others' personalities[43], which can in turn help us predict how they might behave in other contexts. More complex and hierarchical versions of our model might help to illuminate how one's own preferences —or, more broadly, one's self-image—influence these types of social inference. Another open question is whether the biases we observe in both social and non-social groups are sensitive to the level of noise in the environment. Our task used a stable level of stochasticity—correct answers being indicated with a probability of 0.8—while many natural reward environments contain volatility, meaning that the underlying value of a stimulus changes over time[44–46]. In these cases, the extent to which *a priori* stimulus values influence responding—and the advantage this strategy confers—might be different. Lastly, future work might also investigate whether the order and speed at which different stimulus associations are processed affect behavior. In a recent study, Sullivan et al.[47] found that the relative speed at which participants processed the taste versus health properties of food stimuli affected how they chose between junk food and healthful snacks. In a similar vein, participants in our experiments may have differed in how quickly they accessed *a priori* stimulus values (how much they liked the pictured food) relative to their learned values (how likely it is to be the correct answer based on the feedback). Further behavioral and neuroimaging research could help specify in greater detail the temporal dynamics of the decision process.

In conclusion, our study offers a detailed computational account of how we infer and learn others' preferences in the face of uncertainty, showing how our own preferences influence this process at the levels of both learning and choice. We build on prior work showing that humans use their own preferences as priors when learning those of others[10, 11], offering three new contributions: (1) we show how these egocentric influences affect both learning and choice processes, with the latter resulting in a bias that persists in the face of countervailing evidence; (2) we show that these influences are domain-general features of learning, but that priors in social preference learning are specifically sensitive to knowledge about preference prevalence; and (3) we demonstrate that these domain-general, egocentric influences tend to improve average performance on our task. As such, rather than being maladaptive, these influences on learning and choice may help facilitate both social interactions and reward learning more generally.

## Methods

**Participants**. Thirty-three participants (21 female, aged 19–51, mean age 26.7, S.D. = 8.3) took part in the social experiment. Two participants were excluded because their bids on each item poorly predicted their preferences in the choice task, yielding a logistic regression coefficient less than one-fifth of the group's average[48]. An additional 35 participants (25 female, aged 18–48, mean age 24.8,

S.D. = 6.7) took part in the non-social experiment, five of whom were excluded under the same pre-established criterion. Note that we aimed to recruit about 35 participants in each experiment after a pilot study comprising 11 participants yielded reliable model-agnostic results (Supplementary Table 23). All participants were paid £25 plus £0.01 per correct answer, minus the cost of a successfully bid snack item.

Participants were asked to fast and drink only water for 3 h before attending the session. This requirement was intended to ensure that participants were hungry and therefore motivated to spend money to obtain a snack, thereby providing an accurate bid for each item[48]. All participants were screened for current or past use of psychotropic medication, current psychiatric or neurological disorders, diabetes, hypoglycemia, hyperglycemia, and conditions for which fasting up to 5 ½ h would pose a risk. Participants were also screened for proficiency in English and a minimum of about 5 years residency in the UK, to increase the chance of familiarity with the snack stimuli. All participants provided written informed consent. The protocol was approved by the Cambridge Psychology Research Ethics Committee.

**Preference measurement**. Participants were first asked to complete a questionnaire about their familiarity with the snack stimuli and how often they consumed each snack. They then received instructions for the bidding and choice tasks (see Supplementary Methods). Participants were then administered a computerized Becker–Degroot–Marschak bidding procedure[49], in which they were asked to indicate the maximum they were willing to pay, between £0 and £3.00, for each of 40 snack items. These 40 items were then presented in 20 unique pairs, and participants were asked which item in each pair they would prefer to eat. The pairings had previously been determined randomly by computer and were the same for all participants (see Supplementary Table 19 for item pairs; assignment code and output are available on GitHub; see Data availability). Each pair was presented twice, left-right counterbalanced. The order of the pair presentations was random, with the constraint that no pair was presented twice in a row. Participants were told that, at the end of the session, one of their choices would be picked at random by the computer, and a price would be assigned at random between £0.01 and £3.00. If their bid for that item was equal to or above the randomly assigned price, they would receive the item for that price at the end of the session, and the price would be deducted from their payment. If their bid was below the price, they would not receive the item. Participants were told that they would be asked to wait for 1 h at the conclusion of the experiment, during which time they could eat only a purchased item. In reality, we asked participants to stay the balance of their 2 ½ h session (social mean 31.3 min, ranging from 0 to 55 min; non-social: mean 48.3 min, ranging from 29 to 60 min).

**Learning task**. After the bid and choice tasks, participants received instructions for the learning task (see Supplementary Methods). Participants in the social experiment were told that they were learning a set of choices made by a participant in an earlier phase in the study. They were told that this other participant had indicated their preferences between the same pairs of items that they had just seen in the choice task, and that they had to learn which item in each pair the other person had chosen. By contrast, participants in the non-social experiment were told that they were learning a random set of snack items, and that this set of items included one item from each pair they had seen in the choice task. In reality, the sets of items learned by participants in the non-social experiment were also choices made by other people, although this was not told to them.

To generate partner preference data for the learning task, 12 participants (nine female, aged 19–37, mean age 25.8, S.D. = 5.1) took part in a pilot version of the study, which included the choice task described above. Pilot participants' choices were then used to determine the correct and incorrect answers in the learning task for both social and non-social experiments. When a pilot participant's choices were inconsistent (that is, when a participant chose one item during the pair's first presentation and the other item during the second), the choice during the first presentation was used. One pilot participant made perfectly inconsistent choices and was therefore excluded from the partner data, leaving a set of 11 participants' choices. One of these choice sets was then selected at random for each participant in both the social and non-social experiments to learn. Some items in the pilot phase were substituted with similar items in the principal phase due to changes in item availability, so that not all choices made by the pilot participants were exactly the same as the ones learned by the participants. (A full list of item substitutions is available on GitHub; see Data availability.) However, all pairs presented in the learning task were the same pairs presented in the choice task.

Participants were instructed that they would see a yellow feedback box after each response and that it would indicate the correct answer with an 80% probability and the wrong answer with a 20% probability. First, participants completed 14 practice trials using a different set of snack stimuli. This was followed by the main experiment, in which each of the 20 item pairs was presented 30 times for a total of 600 trials. These were divided into three blocks of 200 trials with rest breaks in between blocks. The order of the pairs was pseudorandom, but no pair was presented twice in a row and each pair was presented 10 times within each block. Pairs were left-right counterbalanced so that each item was presented on

each side of the screen five times during each block. Participants were not informed of the experiment's purpose until the debriefing at the conclusion of the session.

**Bayesian learning models**. Bayesian models assumed optimal integration of feedback given specified priors. Over the course of several trials, participants observe ever more feedback indicating the correct item, and the probability of the correct item being inferable as correct will increase relative to the probability of the incorrect item being correct. We modeled this process using Bayes's rule

$$P(A|data_{n+1}) = \frac{P(data_n|A)P(A|data_{n=1})}{P(data_n|A)P(A|data_n) + P(data_n|B)(1 - P(A|data_n))} \quad (6)$$

where $A$ is the correct item, $B$ is the incorrect item and $n$ is the trial number for that particular item pair. $P(A|data)$ is the probability that item $A$ is correct given the feedback presented for that item pair. For trial $n = 1$, this value is the participant's prior for that item pair. $P(A|data)$ is the probability of having seen the observed set of feedback if item $A$ were the correct answer. We model $P(data|A)$ and $P(data|B)$ as binomial functions

$$P(data_n|A) = \frac{n!}{x!(n-x)!} 0.8^x 0.2^{n-x}, \ x = \sum_{i=1}^{n} feedback_{n=i} \quad (7)$$

$$P(data_n|B) = \frac{n!}{x!(n-x)!} 0.2^x 0.8^{n-x}, \ x = \sum_{i=1}^{n} feedback_{n=i} \quad (8)$$

in which $A$ is the correct item, $B$ is the incorrect item, $x$ is the number of times the feedback box has indicated that item on previous trials, and the probabilities correspond to the probabilities of the feedback box in the task indicating the correct answer (80%) versus the wrong answer (20%). Participants were told these probabilities before beginning the task (see Supplementary Methods for full task instructions).

**Drift diffusion models**. Our DDM models were fitted to maximize the likelihood of the observed choices and response times using the method described in[50], which was implemented in Stan[20, 51]. The probability density distributions of response times were calculated as each of four parameters—the threshold distance, drift starting point bias, drift rate and non-decision time—were sampled. Each parameter combination generated two probability density functions: one for an upper threshold response and one for a lower threshold response. The cumulative density of each function is equal to the likelihood of an upper or lower threshold response. In this way, the fitting procedure took account of both response times and choice data. Adjustments to threshold, bias and drift-rate parameters cause different changes to error rates relative to response time distributions[16, 20, 50–52] (Supplementary Fig. 6). For example, a DDM with a high response threshold and a low drift rate has wider distributions with higher means compared to an equally accurate DDM with a low threshold and a high drift rate. In our specification, only DDMs with biased starting points have different distributions for errors than for correct responses. On average, a starting point biased toward the correct threshold results in slower errors than correct responses and vice versa[16]. For this reason, asymmetry in correct versus incorrect response time distributions indicates a biased DDM starting point.

**Parameter estimation**. Mixed-effects regression analyses were performed using version 1.1-8 of the lme4 package[53] for R version 3.2.1[54]. The permutation test was performed using version 1.0.0.0 of the perm package[55] for R. We estimated computational model parameters using the No U-turn Sampling algorithm implemented in Stan[20]. These parameters were estimated hierarchically, which assumed that each participant's parameters were distributed according to group-level means and standard deviations, and with priors similar to those typically used in the literature[56, 57] (see Fig. 5 and Supplementary Table 24 for additional model specification detail). This estimates a group tendency while constraining outliers and allowing for natural variation between participants. The exception was non-decision times, which were not estimated hierarchically in the model. Doing so allowed us to reduce the number of free hyperparameters since participant-level estimates of non-decision time are naturally well constrained by minimum response times. Predictive accuracy was estimated using leave-one-out cross-validation, implemented by the loo package[21] for R, which generated estimates of each model's expected log pointwise predictive density for a new dataset. This method naturally protects against overfitting by indicating which model would likely provide the best predictions for data collected outside of the sample. This method is preferred over other hierarchical model fit indices such as the Deviance Information Criterion because it offers a fully Bayesian estimate of the predictive accuracy of the model, providing us with a measure of uncertainty around this estimate[22]. ELPDs were compared using two-tailed $t$-tests to determine the best model. Full model code is available on GitHub (see Data availability).

**Data availability**. Data, stimuli and full task and analysis code are available on GitHub (www.github.com/bdmlab/ or www.github.com/tortarantola/prior-preferences/) and on Figshare (DOI: 10.6084/m9.figshare.5198572).

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

## Acknowledgements

We thank T. Folke for suggesting the inclusion of item popularity data in our analyses. T.T. was supported by a scholarship from the Cambridge Commonwealth, European, and International Trust; P.D. by the Gatsby Charitable Foundation; and B.D.M. by a Sir Henry Dale Fellowship (no. 102612/A/13/Z), awarded by the Wellcome Trust and the Royal Society.

## Author contributions

T.T. and B.D.M. conceived the study. T.T., B.D.M. and D.K. designed the task. T.T. collected the data. T.T., P.D. and B.D.M. developed the models. T.T. analyzed the data and implemented the models. T.T. prepared the manuscript. T.T., D.K., P.D. and B.D.M. reviewed and edited the manuscript.

## Additional information

**Competing interests:** The authors declare no competing financial interests.

