## [Peer Review File · Nature Communications]

Reviewers' comments:

Reviewer #1 (Remarks to the Author):

The authors demonstrate empirically that most people ground estimates of others' preferences (here: food) in their own preferences. When given a chance to learn about others' preferences they make adjustments in their estimates, but these estimates remain egocentrically biased. These phenomena have been known for a while, but the authors presented several intriguing innovations that makes this paper an interesting advance. Experimentally, their work moves from static social perception to multi-trial learning, an innovation that yields greater inferential power and that allows the authors to try several computational models to represent the findings. I am intrigued by this work, but have to leave to to experts of computational modeling to comment on the soundness of the methods used here. If their soundness can be confirmed, it is my opinion that this work presents a significant advance over extant work in social cognition. Earlier work is being replicated at the phenomenal level, but the contribution lies into opening new ways to think about these phenomena, and thereby opportunities for further research. The authors may want to comment on some of these future directions more explicitly.

To raise one methodological point, I note that the choice of .8 and .2 probability of others' choices being consistent or inconsistent with own preference is arbitrary and it is not varied as a parameter. I presume that such variation is one of the options for future research.

Reviewer #2 (Remarks to the Author):

The authors investigated the question about how our own preferences affect the ability to infer the preferences of others. They investigated how participants learned the food preferences of others. They elicited first individual preferences over a set of food items with a BDM and binary choices and then asked them to learn the choice of others over the same pairs of items. Results show that initial learning was extremely influenced by own preferences, this effect persisted (in terms of higher error rate) in late trials when choices reflected strong differences between own and others' preferences. A non-social version of the task showed how this effect is not domain specific.

Three models of intertrials vs. intra-trial (DDM) computations are reported. The "influenced prior model" has a neutral starting point, and a prior which affects the drift rate computed as a softmax of the relative preferences (i.e. bids) of the participants for the correct vs. the incorrect answer. The "influenced choice model" has a biased DDM starting point modeled as a softmax function of the relative preferences of the participants. And a "dual influence model" with both prior and choices biased by the participants' preferences. Model fitting showed that the dual influence model outperform the other two models and a neutral model (i.e. no effect of own preferences). Moreover, the inclusion of an item popularity shows an improvement in the social condition and no effect on the non-social condition. Simulation analyses confirmed the advantage of the dual influence model.

There are many methodological details that are not reported or are not clear. For instance details about how pairs of items (choice sets) are generated and about the instructions in the non-social task are missing. For instance you never mentioned in the text how you selected the target choice. The only way to understand that indeed only one item was displayed at the time, is watching Fig 1, which is in itself very confusing. Details about the second non-social experiment are missing. how was the non social target programmed to choose? Which kind of instructions the participants got? The method section and the structure of the paper need a serious revision.

My impression is that the authors underestimated the singularity of their experimental paradigm and how this paradigm and instructions could have tremendously affected actual data, thus limiting their generalizability.

Participants are asked to guess the choice of another agent over 20 pairs of food items (i.e. same category but multi-attribute items) repeated 30 times, thus for 600 trials. Participants were instructed that feedback over those choices was stochastic (i.e. correct only 80% of the times).

This is a quite specific learning paradigm. From one side it might limit the dimensionality of the problem (that is, the comparison of any combination of attributes for each combination of items) considering a limited number of items and choice sets, and probably avoided a potential activation of complex analogical reasoning between items' attributes in different trials. In this setting, a possible (sophisticated) strategy of learning the preferences of others would be to try to learn the hierarchical (Lexicographic) structure of those preferences: "I might learn that you like chocolate the most, and would always choose the item with the maximum amount of chocolate on it".

Now assuming, as you have done in your manuscript, that the participants would not use a strategy of this kind, then the task would potentially be reduced to a memory task. I need to learn/memorize how the other chooses over twenty pairs of items and keep guessing always the same for the next 30 repetitions of each pairs. In this case, several simple factors might have affected learning, such as the proximity between one choice set and its reoccurrence (thus requiring more memory as the distance increases); the effect of a wrong feedback on subsequent choice etc. Those effects should be tested and included in your analyses.

The behavioral findings in the learning task are not surprising: participants used their own preferences as priors for the preferences of the others and used common sense popularity index of each item to weight those priors. A simple model of bounded memory and reduction of computational costs could easily explain such a behavioral strategy.

Going back to the computational analyses you presented, I found it of great interest and absolutely timely. You made a serious effort to combine inter-trial learning and intra-trial accumulation of evidence mechanisms. Possible extension to your model would be a consideration of sequential aspects within the intra-trial computations, thus distinguishing a potential first (more basic) stage of evidence acquisition which might be more affected by own preferences and a subsequent (more abstract) stage attribution of preference to the

other. See for instance the work by Sullivan et al 2015.

Reviewer #3 (Remarks to the Author):

This solid paper explores the extent to which one's own preferences (for food) can be used to accurately explore preferences of another person. People are quite good at the task, and there is an initial advantage of predicting other's food choices when they are labeled as a person's choices (rather than unlabeled and hence, "non social"). The computational structure and analysis is state-of-the-art. There are a lot of moving parts so the paper takes some work to grasp, but that is largely a function of the depth of analysis.

The most interesting finding is that including one's own preference as a starting point in DDM can be an advantage when there is noise in observed choice by others (e.g. Figure 6). Intuitively, this is like the classic effect in psychometrics where averaging noisy measures helps, because the target individual's preference is akin to a noisy measure of what others want too.

One limit of the paper in its current form is that it was hard to get a feeling for how overlapping preferences are in this group (though a measure is used in the analysis). Obviously, the degree of preference overlap is an important factor because it will qualify the extent of egocentric advantage and provides a lot of empirical opportunity to push this analysis to other domains. A lot of the early motivating examples about false consensus came, on purpose, from choice domains in which there is a substantial difference in preferences (e.g. the Ross "would you wear a sign" study), and in which there might be few ecological sources of evidence about whether other people share preferences. Other examples, like political attitudes and voting behavior, are in between because there are sharp differences in preferences and some information about how one's own taste might not be shared, but there are also "echo chamber effects" which could limit access to ideal information.

In any case, a simple analysis (perhaps just for Supplementary Information) is to divide subject-food trials into those which are highly shared and not so highly shared and examine performance in both. One plausible divide in your stimulus set is salty and sweet snacks, for which there is apparently some evidence of categorical preferences.

Other comments:

1. The title is too cryptic—it could refer to any number of different studies. Please alter the title so it tells readers more about the paper's contents.
2. ELPD measure is a good one.
3. Supp Fig 1 is very good.
4. Another relevant citation on neural circuitry of 'shopping for somebody else is Janowski et al SCAN 2013

<https://www.ncbi.nlm.nih.gov/pubmed/22349798>

5. Here is another relevant paper on false consensus which is important: Providing information on others wipes out false consensus, but there is some egocentric bias when that information is costly

<http://www.sciencedirect.com/science/article/pii/S0899825612001157>

Reviewer #4 (Remarks to the Author):

Tarantola et al. provided a computational account of how people learn others' preferences for food items, by conducting two behavioral experiments together with computational modeling. Remarkably, the authors combined Bayesian learning framework with Drift Diffusion Model in order to model both inter-trial learning process and intra-trial prediction/choice process. They demonstrated that (i) across trials participants learned the target partner's preference in a manner consistent with a Bayesian learning, in which they used their own preference as a prior belief, and that (ii) within trial the learned preference modulated their prediction behavior by manipulating both the start point and the drift rate in evidence accumulation processes. While I believe the authors addressed the important issue with the solid experiments and data analyses, I have a couple of major concerns.

[Major issues]

1. First of all, the authors should clarify the novelty of this study. They claim the present study provides the first computational account of how one's own preferences interact with learning and predictions about what others value. However, to my knowledge, at least one study (Suzuki et al., PNAS, 2016) demonstrated, by employing Bayesian learning framework, that people use their own preference as a start point of the learning about others' preference (although they did not model intra-trial process). I believe the present study has substantial novelty, especially in terms of modeling both inter-trial and intra-trial processes; but the authors should carefully discuss what is novel in this study.

2. The authors observed that the influence of participants' own preference on the learning process was present also in the non-social control experiment. I am wondering how we should interpret the result. In some places (e.g., abstract) the authors claimed that a domain-general process underlies learning in the two cases. On the other hand, in other places (e.g., page 5), they said distinct forms of learning are engaged in the social and the non-social cases. It would be interesting to discuss the possible psychological process underlying the phenomena.

3. In relation to the 2nd issue, in pages 4-5, the authors provided additional results of the follow-up analyses. I couldn't get the points of these results. Structure of the descriptions could be re-organized.

4. In page 11, the authors claimed that participants' own preferences were predictive of their targets' preferences. A straightforward way to test this is to examine the correlations of food preferences across participants.

5. It would be interesting to discuss the nature of DDM. For instance, I am wondering if biases in the starting point and the drift rate have orthogonal effects. To my eyes, effects of the two factors look qualitatively indifferent. I am wondering what kinds of behavior can be captured by the shift of the starting point but not by the change of the drift rate, and vice versa. Furthermore, I am interested in how biases in the starting point and the drift rate are related to the shift in the decision boundary.

[Minor issues]

1. It would be great if the authors provide detailed information about DDM in Methods.
2. The authors should provide detailed information about model fitting procedure. Especially, although I can understand the hierarchical structure of the parameters (Fig. S1), I don't understand how to compute log likelihood given the set of parameters. As in the previous studies using DDM, likelihood is defined on the joint distribution of choice and RT?
3. The data shown in Fig. 2 are from one example participant? Or, the data are aggregated over the participants? Why not showing error-bars?

Responses to Reviewers' Comments

We thank the four reviewers for assessing our manuscript and for their constructive feedback. We report our responses (in bold) below each comment. We highlight in red the relevant changes in the main text and supplementary materials. For the reviewers' convenience, pertinent changes are also copied below.

Reviewer 1

The authors demonstrate empirically that most people ground estimates of others' preferences (here: food) in their own preferences. When given a chance to learn about others' preferences they make adjustments in their estimates, but these estimates remain egocentrically biased. These phenomena have been known for a while, but the authors presented several intriguing innovations that makes this paper an interesting advance. Experimentally, their work moves from static social perception to multi-trial learning, an innovation that yields greater inferential power and that allows the authors to try several computational models to represent the findings. I am intrigued by this work, but have to leave to to experts of computational modeling to comment on the soundness of the methods used here. If their soundness can be confirmed, it is my opinion that this work presents a significant advance over extant work in social cognition. Earlier work is being replicated at the phenomenal level, but the contribution lies into opening new ways to think about these phenomena, and thereby opportunities for further research. The authors may want to comment on some of these future directions more explicitly.

To raise one methodological point, I note that the choice of .8 and .2 probability of others' choices being consistent or inconsistent with own preference is arbitrary and it is not varied as a parameter. I presume that such variation is one of the options for future research.

Thank you very much for these comments. We agree that an interesting follow-up study might examine how changing the reliability of feedback affects the results we observe. In our simulations, we changed the probability of receiving correct feedback (from 0.8 to 0.7 and 0.6; see Supplementary Fig. 6). We found that, when the reliability of the feedback decreases, the influences of one's own preferences becomes even more advantageous. Following this suggestion, we added three new paragraphs to the Discussion noting possible future directions:

Line 363:

While our results offer a starting point for understanding the dynamics of how we learn about one another's preferences, they also point to several avenues for further research. For instance, like many studies in cognitive neuroscience that investigate the link between learning and social cognition (e.g., ^{6,18,21,39-41}), we purposely used a simple learning paradigm. This allowed us to isolate the effects of one's own preferences on both the learning and choice processes, as well as to dissociate the social nature of preference learning from the more domain-general, Pavlovian effects. While our paradigm was designed to be somewhat naturalistic—we often make discrete choices on behalf of our friends, such as deciding whether to order them a Coke or a Sprite after having watched them make the same choice in the past—learning about others can also involve added layers of complexity. For example, we might use certain types of preference information to infer more general features of others' personalities (e.g., drivers of convertibles are more likely to be extroverted). We might also use more general models of other people to predict how they might behave in specific contexts (e.g., a health-conscious friend will be more likely to prefer a salad to a hamburger). More complex and hierarchical versions of our model might help to illuminate how one's own preferences—or, more broadly, one's own self-conception—influence these types of social inference.

Our domain-general results also suggest several possible elaborations to investigate reinforcement learning dynamics more broadly. We noted that, even without the benefit of insight into others' preferences, our participants were remarkably well-tuned in terms of the level of bias that would maximize their performance in the task. One open question is whether the extent of this bias is sensitive to the level of noise in the environment. Our task used a stable level of stochasticity—correct answers being indicated with an 0.8 probability—while many natural reward environments contain volatility, meaning that the underlying value of a

stimulus changes over time⁴²⁻⁴⁴. In these cases, the extent to which *a priori* stimulus values influence responding—and the advantage this strategy confers—might be different.

Lastly, future work might also investigate whether the order and speed at which different stimulus associations are processed affect behavior. In a recent study, Sullivan and colleagues⁴⁵ found that the relative speed at which participants processed the taste versus health properties of food stimuli affected how they chose between junk food and healthful snacks. In a similar vein, participants in our experiments may have differed in how quickly they accessed *a priori* stimulus values (how much they liked the pictured food) relative to their learned values (how likely it is to be the correct answer based on the feedback). Further behavioral and neuroimaging research could help specify in greater detail the temporal dynamics of the decision process.

Reviewer 2

The authors investigated the question about how our own preferences affect the ability to infer the preferences of others. They investigated how participants learned the food preferences of others. They elicited first individual preferences over a set of food items with a BDM and binary choices and then asked them to learn the choice of others over the same pairs of items. Results show that initial learning was extremely influenced by own preferences, this effect persisted (in terms of higher error rate) in late trials when choices reflected strong differences between own and others' preferences. A non-social version of the task showed how this effect is not domain specific.

Three models of intertrials vs. intra-trial (DDM) computations are reported. The “influenced prior model” has a neutral starting point, and a prior which affects the drift rate computed as a softmax of the relative preferences (i.e. bids) of the participants for the correct vs. the incorrect answer. The “influenced choice model” has a biased DDM starting point modeled as a softmax function of the relative preferences of the participants. And a “dual influence model” with both prior and choices biased by the participants' preferences. Model fitting showed that the dual influence model outperform the other two models and a neutral model (i.e. no effect of own preferences). Moreover, the inclusion of an item popularity shows an improvement in the social condition and no effect on the non-social condition. Simulation analyses confirmed the advantage of the dual influence model.

There are many methodological details that are not reported or are not clear. For instance details about how pairs of items (choice sets) are generated and about the instructions in the non-social task are missing. For instance you never mentioned in the text how you selected the target choice. The only way to understand that indeed only one item was displayed at the time, is watching Fig 1, which is in itself very confusing. Details about the second non-social experiment are missing. how was the non social target programmed to choose? Which kind of instructions the participants got? The method section and the structure of the paper need a serious revision.

Thank you for highlighting these issues—we apologize for the confusing and incomplete descriptions of our methods. We have substantially revised the Introduction, Results, and Methods to clarify how the items were paired and how the target choices were collected and assigned. We also included the full task instructions as Supplementary Texts 1 and 2. In addition, we redrew Figure 1 in order to make it clear that the targets' preference sets were generated by their choices between the item pairs. (The new figure is copied below.)

Line 49:

However, even after learning had plateaued, participants continued to make more errors when expressing choices that differed from their own preferences. In a non-social follow-up version of the experiment, **a different group of participants performed exactly the same task, but received different instructions. This second study showed the egocentric influence we had identified to be domain-general and potentially applicable to reward learning more broadly. However, we also identified a key distinction between the social and non-social groups: participants in the social experiment used some additional insight into the popularity of snack items, which improved their performance.**

Line 100:**Social vs. Non-social Framing**

In order to determine whether the effect we observed was specific to the social nature of the task, we ran a second, non-social experiment in which participants performed the same task but received different instructions. In the non-social instructions, participants were asked to learn the correct item in each of the same 20 pairs of snacks. Unlike in the social experiment, these participants were not informed that the set of items they were learning were choices made by another person (see Supplementary Texts 1 and 2 for instructions). Other than the difference in instructions, the two tasks were the same. As in the social experiment, participants in the non-social experiment were actually learning choices made by a randomly chosen pilot participant. Unlike in the social experiment, however, this was not told to them.

Line 433:

Preference measurement. Participants were first asked to complete a questionnaire about their familiarity with the snack stimuli and how often they consumed each snack. They then received instructions for the bidding and choice tasks (Supplementary Texts 1 and 2). Participants were then administered a computerized Becker-DeGroot-Marschak (BDM) bidding procedure⁴⁷, in which they were asked to indicate the maximum they were willing to pay, between £0 and £3.00, for each of 40 snack items. These 40 items were then presented in 20 unique pairs, and participants were asked which item in each pair they would prefer to eat. The pairings had previously been determined randomly by computer and were the same for all participants (see Supplementary Table 6 for item pairs; assignment code and output are available on GitHub). Each pair was presented twice, left-right counterbalanced. The order of the pair presentations was random, with the constraint that no pair was presented twice in a row.

Line 453:

Learning task. After the bid and choice tasks, participants received instructions for the learning task (see Supplementary Texts 1 and 2). Participants in the social experiment were told that they were learning a set of choices made by a participant in an earlier phase in the study. They were told that this other participant had indicated their preferences between the same pairs of items that they had just seen in the choice task, and that they had to learn which item in each pair the other person had chosen. By contrast, participants in the non-social experiment were told that they were learning a random set of snack items, and that this set of items included one item from each pair they had seen in the choice task. In reality, the sets of items learned by participants in the non-social experiment were also choices made by other people, though this was not told to them.

To generate target preference data for the learning task, 12 participants (9 female, aged 19-37, mean age 25.8) took part in a pilot version of the study, which included the choice task described above. Pilot participants' choices were then used to determine the correct and incorrect answers in the learning task for both social and non-social experiments. When a pilot participant's choices were inconsistent (that is, when a participant chose one item during the pair's first presentation and the other item during the second), the choice during the first presentation was used. One pilot participant made perfectly inconsistent choices and was therefore excluded from the target data, leaving a set of 11 participants' choices. One of these choice sets was then selected at random for each participant in both the social and non-social experiments to learn.

Line 479:

The main phase was preceded by 14 practice trials, which used a different set of snack stimuli. In the main phase, each of the 20 item pairs was presented 30 times for a total of 600 trials, which were divided into three blocks of 200 trials with rest breaks in between. The order of the pairs was random, with the constraints that no pair was presented twice in a row and each pair was presented 10 times within each block. Pairs were left-right counterbalanced so that each item was presented on each side of the screen 5 times during each block.

Figure 1. Learning task. Participants were asked to indicate which choice they believed the target made. After making a response, a yellow feedback box indicated the correct answer with 80% probability. Participants saw 20 different pairs, interleaved, 30 times each for a total of 600 trials.

My impression is that the authors underestimated the singularity of their experimental paradigm and how this paradigm and instructions could have tremendously affected actual data, thus limiting their generalizability.

Participants are asked to guess the choice of another agent over 20 pairs of food items (i.e. same category but multi-attribute items) repeated 30 times, thus for 600 trials. Participants were instructed that feedback over those choices was stochastic (i.e. correct only 80% of the times).

This is a quite specific learning paradigm. From one side it might limit the dimensionality of the problem (that is, the comparison of any combination of attributes for each combination of items) considering a limited number of items and choice sets, and probably avoided a potential activation of complex analogical reasoning between items' attributes in different trials. In this setting, a possible (sophisticated) strategy of learning the preferences of others would be to try to learn the hierarchical (Lexicographic) structure of those preferences: "I might learn that you like chocolate the most, and would always choose the item with the maximum amount of chocolate on it".

Thank you for these thoughtful comments. As you correctly note, we adapted a learning paradigm—similar to those commonly used in related studies (Behrens et al., 2009, 2008; Garvert et al., 2015; Kumaran et al., 2016; Lin et al., 2012; Suzuki et al., 2016, 2012)—with the aim of isolating the effects of one's own preferences on both the learning and choice processes, as well as to dissociate the effect of the social framing from the more domain-general, Pavlovian effects. Notably, using a very different design, one of these studies (Suzuki et al., 2016) reports a similar egocentric effect on participants' priors when learning risk preferences.

We have also revised the manuscript to discuss the important point you raise, namely that learning about others can also benefit from more complex forms of social reasoning. While our paradigm was designed to be somewhat naturalistic—we often make discrete choices on behalf of our friends, such as deciding whether to order them a Coke or a Sprite after having watched them make the same choice in the past—learning about others can also involve added layers of complexity. This might include building models of others' preferences based on observing discrete choices, as well as predicting discrete choices based on these more general models.

Line 363:

While our results offer a starting point for understanding the dynamics of how we learn about one another's preferences, they also point to several avenues for further research. For instance, like many studies in cognitive neuroscience that investigate the link between learning and social cognition (e.g., ^{6,18,21,39-41}), we purposely used a simple learning paradigm. This allowed us to isolate the effects of one's own preferences on both the learning and choice processes, as well as to dissociate the social nature of preference learning from the more domain-general, Pavlovian effects. While our paradigm was designed to be somewhat naturalistic—we often make discrete choices on behalf of our friends, such as deciding whether to order them a Coke or a Sprite after having watched them make the same choice in the past—learning about others can also involve added layers of complexity. For example, we might use certain types of preference information to infer more general features of others' personalities (e.g., drivers of convertibles are more likely to be extroverted). We might also use more general models of other people to predict how they might behave in specific contexts (e.g., a health-conscious friend will be more likely to prefer a salad to a hamburger). More complex and hierarchical versions of our model might help to illuminate how one's own preferences—or, more broadly, one's own self-conception—influence these types of social inference.

Our domain-general results also suggest several possible elaborations to investigate reinforcement learning dynamics more broadly. We noted that, even without the benefit of insight into others' preferences, our participants were remarkably well-tuned in terms of the level of bias that would maximize their performance in the task. One open question is whether the extent of this bias is sensitive to the level of noise in the environment. Our task used a stable level of stochasticity—correct answers being indicated with an 0.8 probability—while many natural reward environments contain volatility, meaning that the underlying value of a stimulus changes over time⁴²⁻⁴⁴. In these cases, the extent to which *a priori* stimulus values influence responding—and the advantage this strategy confers—might be different.

More generally, we are sorry that our original description of the social versus non-social results was unclear. We revised the manuscript to clarify that the task performed by the participants in the two experiments was exactly the same—including the sources from which the correct and incorrect answers were drawn. The difference is that the two groups received different instructions (now included as Supplementary Texts 1 and 2). These different instructions elicited one critical difference in the results: participants who received the “social” instructions included item popularity in their priors, while the participants who received the “non-social” instructions did not. We believe this difference makes a strong case for the generalizability of our results', because it shows that the process of specifically social reasoning was independent of the task itself. Regrettably, our initial descriptions prevented this from being clear.

Line 100:

Social vs. Non-social Framing

In order to determine whether the effect we observed was specific to the social nature of the task, we ran a second, non-social experiment in which participants performed the same task but received different instructions. In the non-social instructions, participants were asked to learn the correct item in each of the same 20 pairs of snacks. Unlike in the social experiment, these participants were not informed that the set of items they were learning were choices made by another person (see Supplementary Texts 1 and 2 for instructions). Other than the difference in instructions, the two tasks were the same. As in the social experiment, participants in the non-social experiment were actually learning choices made by a randomly chosen pilot participant. Unlike in the social experiment, however, this was not told to them.

Line 120:

To test this hypothesis, we included an index of item popularity in our models, measured by the percentage of other participants in both experiments who bid more for an item compared to its alternative (Supplementary Table 6). (The level of agreement varied among item pairs, from 80% of participants bidding more for Maltesers than Sour Patch Kids, to being about evenly split between the Bounty bar and sweetcorn; the mean level of agreement was 69.8%.)

Including item popularity significantly improved a logistic model of first trial performance in the social group ($\chi^2(1)=38.3$, $p<10^{-8}$ corrected) and, to a lesser extent, in the non-social group ($\chi^2(1)=8.1$, $p<0.01$ corrected). This effect was significantly greater for the social group than the non-social group ($\chi^2(1)=5.7$, $p<0.02$; Supplementary Table 7; illustrated in Fig. 3 and Supplementary Fig. 1).

Now assuming, as you have done in your manuscript, that the participants would not use a strategy of this kind, then the task would potentially be reduced to a memory task. I need to learn/memorize how the other chooses over twenty pairs of items and keep guessing always the same for the next 30 repetitions of each pairs. In this case, several simple factors might have affected learning, such as the proximity between one choice set and its reoccurrence (thus requiring more memory as the distance increases); the effect of a wrong feedback on subsequent choice etc. Those effects should be tested and included in your analyses.

Following your suggestions, we have now conducted a number of additional analyses. We repeated our regression analyses, this time controlling for (1) the number of trials since the last presentation of the current item pair, and (2) whether the last feedback for that item pair was accurate. These analyses confirm our initial findings. As an additional check, we also repeated these analyses using an alternate measure of preference congruence—whether the participant indicated that they preferred the same item during the choice task (rather than using their relative responses during the bid task). This further confirmed our results.

Line 91:

We also ran these models using participants' own choices between the items (rather than the differences in their bids for each item, Δv) as a measure of preference congruence. These analyses yielded substantially the same results (Supplementary Tables 3 and 4). Our task pseudorandomized both (1) the number of trials between subsequent presentations of the same item pairs and (2) the accuracy of the feedback presented on any given trial (see Methods). Nevertheless, to rule out potential confounds, our regression models corrected for both these factors. We also ran separate regression models to ensure that neither of these factors was predicted by Δv (Supplementary Table 5).

The behavioral findings in the learning task are not surprising: participants used their own preferences as priors for the preferences of the others and used common sense popularity index of each item to weight those priors. A simple model of bounded memory and reduction of computational costs could easily explain such a behavioral strategy.

Going back to the computational analyses you presented, I found it of great interest and absolutely timely. You made a serious effort to combine inter-trial learning and intra-trial accumulation of evidence mechanisms. Possible extension to your model would be a consideration of sequential aspects within the intra-trial computations, thus distinguishing a potential first (more basic) stage of evidence acquisition which might be more affected by own preferences and a subsequent (more abstract) stage attribution of preference to the other. See for instance the work by Sullivan et al 2015.

Thank you very much for these comments and for drawing our attention to this very interesting paper (which we now describe in the Discussion) and the potential extension of our model. We have updated the Discussion to elaborate on how future work might more specifically isolate the temporal dynamics behind the effects we observe:

Line 388:

Lastly, future work might also investigate whether the order and speed at which different stimulus associations are processed affect behavior. In a recent study, Sullivan and colleagues⁴⁵ found that the relative speed at which participants processed the taste versus health properties of food stimuli affected how they chose between junk food and healthful snacks. In a similar vein, participants in our experiments may have differed in how quickly they accessed *a priori* stimulus values (how much they liked the pictured food) relative

to their learned values (how likely it is to be the correct answer based on the feedback). Further behavioral and neuroimaging research could help specify in greater detail the temporal dynamics of the decision process.

Reviewer 3

This solid paper explores the extent to which one's own preferences (for food) can be used to accurately explore preferences of another person. People are quite good at the task, and there is an initial advantage of predicting other's food choices when they are labeled as a person's choices (rather than unlabeled and hence, "non social"). The computational structure and analysis is state-of-the-art. There are a lot of moving parts so the paper takes some work to grasp, but that is largely a function of the depth of analysis.

Thank you very much for this positive endorsement of our paper.

The most interesting finding is that including one's own preference as a starting point in DDM can be an advantage when there is noise in observed choice by others (e.g. Figure 6). Intuitively, this is like the classic effect in psychometrics where averaging noisy measures helps, because the target individual's preference is akin to a noisy measure of what others want too.

One limit of the paper in its current form is that it was hard to get a feeling for how overlapping preferences are in this group (though a measure is used in the analysis). Obviously, the degree of preference overlap is an important factor because it will qualify the extent of egocentric advantage and provides a lot of empirical opportunity to push this analysis to other domains. A lot of the early motivating examples about false consensus came, on purpose, from choice domains in which there is a substantial difference in preferences (e.g. the Ross "would you wear a sign" study), and in which there might be few ecological sources of evidence about whether other people share preferences. Other examples, like political attitudes and voting behavior, are in between because there are sharp differences in preferences and some information about how one's own taste might not be shared, but there are also "echo chamber effects" which could limit access to ideal information.

Thank you for these insightful suggestions. We have included a table of the item pairs, with summary popularity statistics, as new Supplementary Table 6, and included a brief summary in the main text:

Line 120:

To test this hypothesis, we included an index of item popularity in our models, measured by the percentage of other participants in both experiments who bid more for an item compared to its alternative (Supplementary Table 6). (The level of agreement varied among item pairs, from 80% of participants bidding more for Maltesers than Sour Patch Kids, to being about evenly split between the Bounty bar and sweetcorn; the mean level of agreement was 69.8%.)

Including item popularity significantly improved a logistic model of first trial performance in the social group ($\chi^2(1)=38.3$, $p<10^{-8}$ corrected) and, to a lesser extent, in the non-social group ($\chi^2(1)=8.1$, $p<0.01$ corrected). This effect was significantly greater for the social group than the non-social group ($\chi^2(1)=5.7$, $p<0.02$; Supplementary Table 7; illustrated in Fig. 3 and Supplementary Fig. 1).

In any case, a simple analysis (perhaps just for Supplementary Information) is to divide subject-food trials into those which are highly shared and not so highly shared and examine performance in both.

Following your suggestion, we added a new Supplementary Figure 1, which illustrates these effects separately for choices that were widely shared versus not widely shared. This figure illustrates more clearly the statistical effect we report in the main text, showing a parametric effect of item popularity on first trial performance in the social group. It also shows that the effect of preference congruence maintains both when preferences are widely shared in the population and when there is disagreement.

Supplementary Figure 1. Performance on first trials for each item pair, separated by preference congruence and item popularity. The effect of item popularity on first trial performance in the social group was greater when the popularity of the learned choices was greater (bottom panel) than when choices were less widely shared (top panel; see main text for statistical tests). This shows the parametric effect of item popularity illustrated in Fig. 3 (main text). “More popular” items are those chosen by more than 50% of participants in both experiments; “less popular” items were chosen by fewer than 50%. Data in the top panel represent the 10 item pairs for which the 61 social and non-social participants showed the least agreement in their choices (49-72% making the same choice). Data in the bottom panel represent the other 10 item pairs in which participants showed the most agreement (75-90%). Error bars represent bootstrapped standard errors clustered by participant.

One plausible divide in your stimulus set is salty and sweet snacks, for which there is apparently some evidence of categorical preferences.

Unfortunately, because our item pairs were randomly chosen and fixed across participants, only 50% of them contain a clear choice within the two categorical divisions we identified: between a salty and a sweet option, or between a chocolate and a non-chocolate option. As a result, we found it more prudent to divide our item pairs by how widely shared the choices were in our sample (see above).

Other comments:

1. The title is too cryptic—it could refer to any number of different studies. Please alter the title so it tells readers more about the paper’s contents.

Thank you for this note. We have changed the title of the paper to “Prior preferences beneficially influence social and non-social learning.”

2. ELPD measure is a good one.

3. Supp Fig 1 is very good.

Thank you very much.

4. Another relevant citation on neural circuitry of ‘shopping for somebody else is Janowski et al SCAN 2013
<https://www.ncbi.nlm.nih.gov/pubmed/22349798>

5. Here is another relevant paper on false consensus which is important: Providing information on others wipes out false consensus, but there is some egocentric bias when that information is costly
<http://www.sciencedirect.com/science/article/pii/S0899825612001157>

Thank you for bringing this work to our attention. We have cited both papers in our revised manuscript (refs. 9 and 22):

Reviewer 4

Tarantola et al. provided a computational account of how people learn others’ preferences for food items, by conducting two behavioral experiments together with computational modeling. Remarkably, the authors combined Bayesian learning framework with Drift Diffusion Model in order to model both inter-trial learning process and intra-trial prediction/choice process. They demonstrated that (i) across trials participants learned the target partner’s preference in a manner consistent with a Bayesian learning, in which they used their own preference as a prior belief, and that (ii) within trial the learned preference modulated their prediction behavior by manipulating both the start point and the drift rate in evidence accumulation processes. While I believe the authors addressed the important issue with the solid experiments and data analyses, I have a couple of major concerns.

[Major issues]

1. First of all, the authors should clarify the novelty of this study. They claim the present study provides the first computational account of how one’s own preferences interact with learning and predictions about what others value. However, to my knowledge, at least one study (Suzuki et al., PNAS, 2016) demonstrated, by employing Bayesian learning framework, that people use their own preference as a start point of the learning about others’ preference (although they did not model intra-trial process). I believe the present study has substantial novelty, especially in terms of modeling both inter-trial and intra-trial processes; but the authors should carefully discuss what is novel in this study.

Thank you for bringing this paper’s finding to our attention. We have now cited the paper (ref. 6) and revised the Introduction and Discussion to specify more carefully the original contributions of our paper to the existing literature: namely, we show (1) the separate influences of egocentric bias on learning and choice processes; (2) that these influences are domain-general, but that priors in the social experiment are specifically sensitive to preference prevalence; and (3) that the effects we identify yield performance benefits.

Line 28:

When learning to navigate new environments, it often helps to have some prior information to work from. This is also true in social environments, which require us to learn and predict others’ preferences, often based on limited information. The stakes can be high—such as maintaining interpersonal relationships, operating efficiently in a market, and resolving conflicts between institutions or governments—and starting off on the right track can be important. In such cases, a useful starting point might be our own preferences –

absent evidence to the contrary, it is reasonable to assume that other people prefer the same things that we do. For example, when buying a gift for a friend, we might find ourselves using our own tastes as a guide, especially when we have little information about what the recipient might like. Indeed, it is well-established in psychology that people tend to project their own values, traits, and preferences onto others¹⁻⁴ **and use themselves as priors when learning others' preferences**^{5,6}.

As we gather more information, however, we ought to update our predictions. For instance, our gift buying should be improved as we learn more about our friend's tastes, past purchases, or, if we're lucky, if he drops an occasional hint. Nevertheless, research in psychology has shown that egocentric influences tend to persist, even in the face of countervailing evidence^{5,7,8}, **and especially when seeking out that evidence requires effort**⁹. **Still unclear are the dynamics of how these influences shape both learning and choice processes, whether they are exclusively social or result from domain-general biases, and what the implications are for performance.**

Line 399:

We build on prior work showing that humans use their own preferences as priors when learning those of others^{5,6}, offering three new insights: (1) we show how these egocentric influences affect both learning and choice processes, with the latter resulting in a persistent bias; (2) we show that these influences are domain-general features of learning, but that priors in social preference learning are specifically sensitive to insight into preference prevalence; and (3) we demonstrate that these domain-general, egocentric influences tend to **improve average performance**. **As such, rather than being maladaptive, these influences might provide a beneficial edge by helping to facilitate social interactions and reward learning more generally.**

2. The authors observed that the influence of participants' own preference on the learning process was present also in the non-social control experiment. I am wondering how we should interpret the result. In some places (e.g., abstract) the authors claimed that a domain-general process underlies learning in the two cases. One the other hand, in other places (e.g., page 5), they said distinct forms of learning are engaged in the social and the non-social cases. It would be interesting to discuss the possible psychological process underlying the phenomena.

Thank you for this comment. We apologize that our initial description of the social/non-social manipulation, as well as the differences in their results, were unclear. We have substantially revised our description of these findings:

Line 14:

ABSTRACT

Our own preferences affect a broad array of social behaviors. This includes the way we learn the preferences of others, an ability that often relies on limited or ambiguous information. Here, using a reinforcement learning paradigm, **we describe this egocentric influence on learning, finding it to be** reflected in both performance and response times. Through the medium of computational models that combine inter-trial learning and intra-trial choice, we found both transient effects of participants' preferences on the former through the influence of priors, and persistent effects on the latter. A second experiment showed that these effects **generalize to non-social learning as well, albeit not allowing participants to take advantage of additional insight into the prevalence of different preferences in the social context**. We further found that, rather than being disadvantageous, the **domain-general** egocentric influences we identified can yield performance advantages in uncertain environments.

Line 52

... a different group of participants performed exactly the same task, but received different instructions. This second study showed the egocentric influence we had identified to be domain-general and potentially applicable to reward learning more broadly. However, we also identified a key distinction between the social and non-social groups: participants in the social experiment used some additional insight into the popularity of snack items, which improved their performance.

Line 100:

Social vs. Non-social Framing

In order to determine whether the effect we observed was specific to the social nature of the task, we ran a second, non-social experiment in which participants performed the same task but received different instructions. In the non-social instructions, participants were asked to learn the correct item in each of the same 20 pairs of snacks. Unlike in the social experiment, these participants were not informed that the set of items they were learning were choices made by another person (see Supplementary Texts 1 and 2 for instructions). Other than the difference in instructions, the two tasks were the same. As in the social experiment, participants in the non-social experiment were actually learning choices made by a randomly chosen pilot participant. Unlike in the social experiment, however, this was not told to them.

Line 120:

To test this hypothesis, we included an index of item popularity in our models, measured by the percentage of other participants in both experiments who bid more for an item compared to its alternative (Supplementary Table 6). (The level of agreement varied among item pairs, from 80% of participants bidding more for Maltesers than Sour Patch Kids, to being about evenly split between the Bounty bar and sweetcorn; the mean level of agreement was 69.8%.)

Including item popularity significantly improved a logistic model of first trial performance in the social group ($\chi^2(1)=38.3$, $p<10^{-8}$ corrected) and, to a lesser extent, in the non-social group ($\chi^2(1)=8.1$, $p<0.01$ corrected). This effect was significantly greater for the social group than the non-social group ($\chi^2(1)=5.7$, $p<0.02$; Supplementary Table 7; illustrated in Fig. 3 and Supplementary Fig. 1).

These revisions, we hope, better frame our discussion of the psychology underlying the key difference: namely, that the prior in the “social” group include both the participant’s own preference as well as some insight into the popularity of the choice, while the prior in the “non-social” group only includes the participant’s own preference.

3. In relation to the 2nd issue, in pages 4-5, the authors provided additional results of the follow-up analyses. I couldn’t get the points of these results. Structure of the descriptions could be re-organized.

Thank you for this note. We apologize that our original description was a bit opaque. We have reorganized this section (shown above) and added a new Figure 3, which hopefully make the results clearer.

Figure 3. Performance on first trials for each item pair, separated by preference congruence and item popularity. Performance on first trials, before any feedback had been presented, was substantially affected by participants’ own preferences. Participants in the social experiment (left panel), however, were more sensitive to item popularity, thus improving their performance. (See main text for statistical tests.) “More popular” items are those chosen by more than 50% of participants in both experiments; “less popular”

items were chosen by fewer than 50%. Error bars represent bootstrapped standard errors clustered by participant.

4. In page 11, the authors claimed that participants' own preferences were predictive of their targets' preferences. A straightforward way to test this is to examine the correlations of food preferences across participants.

Thank you for this suggestion. We revised that section to include a correlation statistic. We also include a table of item pairs with summary popularity statistics at our new Supplementary Table 6.

Line 260:

The advantage we observed in our simulations resulted chiefly from the fact that participants' own preferences were, on average, predictive of their targets' preferences. (Δv correlates strongly with the ratio of other participants who prefer the correct item; $r=0.48$, $p<10^{-69}$.)

5. It would be interesting to discuss the nature of DDM. For instance, I am wondering if biases in the starting point and the drift rate have orthogonal effects. To my eyes, effects of the two factors look qualitatively indifferent. I am wondering what kinds of behavior can be captured by the shift of the starting point but not by the change of the drift rate, and vice versa. Furthermore, I am interested in how biases in the starting point and the drift rate are related to the shift in the decision boundary.

Thank you for raising this important detail. We have updated the Methods section to provide a more comprehensive description of the DDM, as well as some qualitative discussion of how the different parameters affect data predictions, with a new Supplementary Figure 7. Specifically, increases in drift rates have symmetric effects on correct versus incorrect response time distributions, while changes in the bias term have asymmetric effects.

Line 510:

Drift diffusion models. Our DDM models were fitted to maximize the likelihood of the observed choices and response times. Using the method described in ⁴⁸ and implemented in Stan ^{15,49}, the probability density distributions of response times were calculated as each of four parameters—the threshold distance, drift starting point bias, drift rate, and non-decision time—were sampled. Each parameter combination generates two probability density functions: one for an upper threshold response and one for a lower threshold response. The cumulative density of each function is equal to the likelihood of an upper or lower threshold response. In this way, the fitting procedure takes account of both response times and choice data. Adjustments to threshold, bias, and drift rate parameters cause different changes to error rates relative to response time distributions ^{15,48,50,49,51} (Supplementary Fig. 7). For example, a DDM with a high response threshold and a low drift rate has wider distributions with higher means compared to an equally accurate DDM with a low threshold and a high drift rate. Only DDMs with biased starting points have different distributions for errors than for correct responses. On average, a starting point biased toward the correct threshold results in slower errors than correct responses and vice versa ⁵¹. For this reason, asymmetry in correct versus incorrect response time distributions indicates a biased DDM starting point.

Supplementary Figure 7. Different effects of drift diffusion model parameters on response time distributions. Each of the three models increases accuracy from about 73% (gray plots) to about 88% by changing different parameters. Increasing drift rates sharpens response time distributions for both correct and incorrect responses (blue; top panel). By contrast, biasing the starting point toward the correct response threshold affects correct and incorrect response time distributions differently (yellow; middle panel). Increasing threshold distance broadens response time distributions evenly for correct and incorrect responses (red; bottom panel). For illustration, plots show the probability density of response times given the response (i.e., not weighted by the likelihood of the response itself). Each model is based on 10,000 simulations using ⁴⁹. Distributions are kernel density smoothed.

[Minor issues]

1. It would be great if the authors provide detailed information about DDM in Methods.

Thank you for this suggestion. We have included some additional detail about the DDM in a new “Drift diffusion models” subsection in Methods, shown above.

2. The authors should provide detailed information about model fitting procedure. Especially, although I can understand the hierarchical structure of the parameters (Fig. S1), I don't understand how to compute log likelihood given the set of parameters. As in the previous studies using DDM, likelihood is defined on the joint distribution of choice and RT?

We have included some additional detail about the DDM and model fitting procedure in the Methods section (shown above). We also clarified that the likelihood is defined by the probability density function of response times given model parameters, but that these functions are weighted by the likelihood of choices, so that the cumulative density of response times for one choice equals the total likelihood of that choice.

3. The data shown in Fig. 2 are from one example participant? Or, the data are aggregated over the participants? Why not showing error-bars?

Thank you for noting this. We revised the Figure 2 caption to clarify that the data are aggregated over all participants. We omitted error bars for clarity of presentation and because each line represents different numbers of trials for each participant, and the interpretation of the error bars would therefore be ambiguous. The discretization between shared and unshared preferences in the plots is for illustration purposes. Unfortunately, we could not work out how to show the statistical measures from the regression models—which use a continuous measure of preference congruence—without causing confusion.

Figure 2. Observed and predicted performance and response times. Average model predictions for the winning *dual influence* model with item popularity (right panels) compared to the data (left panels) **averaged across participants in the social (a) and non-social (b) groups...**

REVIEWERS' COMMENTS:

Reviewer #1 (Remarks to the Author):

In my opinion, the authors addressed the reviewers' concerns and suggestions well and I recommend that this version of the manuscript be published.

Reviewer #2 (Remarks to the Author):

The new version of the manuscript represents a significant improvement compared to the original version. The method is now well described in the manuscript and the analysis way more clear than the previous version. This paper represents a significant improvement for the understanding of social learning.

Reviewer #3 (Remarks to the Author):

This paper was revised very carefully. Thanks to the authors for diligence in doing so.

Reviewer #4 (Remarks to the Author):

The authors have adequately addressed my concerns.